# Multiplexed lipid nanoparticle barcoding reveals tissue-dynamic kinetic insights and enriched cellular tropism in hepatic zones

Stephen T. Moore[1,2], Xizhen Lian[1], Amogh Vaidya [1], Sumanta Chatterjee[1], Julien Santelli[1], Yehui Sun [1], Lukas Farbiak [1], Hao Zhu [2] ✉ & Daniel J. Siegwart [1] ✉

Lipid nanoparticles (LNPs) efficiently deliver nucleic acids to cells in vivo and facilitate clinical applications including RNA-based vaccines and therapies. Discovery and optimization of LNPs remain challenging due to the complexity of input variables and low throughput workflows. To accelerate these processes, we report a broadly compatible barcoded Cre recombinase mRNA barcode platform that enables multiplexed LNP tracking in vivo in tdTomato reporter mice. We evaluate accumulation and degradation kinetics of mRNA encapsulated in Selective Organ Targeting (SORT) LNPs in the liver, lung, and spleen, and show that functional protein activity is associated with rapid organ enrichment. We further demonstrate how barcode multiplexing can streamline systematic kinetic studies, distinguish nanoparticles with distinct biological outcomes, and differentiate subtle, yet important, variations within a series of similar formulations. Finally, we use barcoding to identify and characterize nanoparticles with hepatic zonal bias and previously overlooked extrahepatic tropism. This approach could accelerate high resolution characterization of nanoparticles with desirable properties, enable large-scale systematic studies of diverse LNPs, and provide insights into optimizable parameters of LNP-mRNA delivery.

Lipid nanoparticles (LNPs) have emerged as the most clinically advanced platform for the delivery of nucleic acids for intramuscular RNA vaccines[1] and intravenous (IV) therapies targeting liver hepatocytes[2,3]. Emerging technologies are beginning to enable delivery of mRNA to cells beyond the liver[4], setting the stage for expanded therapeutic applications beyond COVID-19 mRNA vaccines[5]. Indeed, discovery and optimization of LNPs capable of IV delivery of mRNA to specific cells, such as hepatocytes within particular metabolic zones, remain unsolved due to the complexity of input variables and low-throughput workflows. Here we report a barcoded mRNA platform

that enabled streamlined optimization workflows, revealed tissue-specific kinetic insights, and identified LNPs with unique cellular tropisms.

LNPs are formulated through controlled self-assembly of nucleic acids and amphipathic lipids selected from large libraries of available materials[6–8]. Therefore, an immense design space to engineer the physical and chemical properties of LNPs exists through modulation of many factors, including lipid chemistry, stoichiometry, number and class, formulation conditions, and nucleic acid type, among others. Recently, strategies for organ- and cell-enriched LNP delivery have

[1]Department of Biomedical Engineering, Department of Biochemistry, Simmons Comprehensive Cancer Center, Program in Genetic Drug Engineering, The University of Texas Southwestern Medical Center, Dallas, TX, USA. [2]Children's Research Institute, Departments of Pediatrics and Internal Medicine, Center for Regenerative Science and Medicine, Children's Research Institute Mouse Genome Engineering Core, University of Texas Southwestern Medical Center, Dallas, TX, USA. ✉e-mail: Hao.Zhu@UTSouthwestern.edu; Daniel.Siegwart@UTSouthwestern.edu

emerged, including endogenous and active targeting mechanisms that involve the use of novel lipid chemistries and LNP surface functionalization[4,9,10]. For example, we and others have shown that Selective ORgan Targeting (SORT) can enable RNA delivery to the liver, lung, spleen, and bone marrow, through the addition of supplemental lipid species to LNP formulations[11–16]. Despite these major advances in targeting and delivery, current culture models and workflows for LNP tropism investigation, including organ imaging, reporter animals, and cell surface marker identification, do not fully capture relevant biological fates of LNPs and can potentially miss evidence of delivery in currently unreachable tissues. Additionally, there are major knowledge gaps in the pharmacological profiles of LNP fates that could be elucidated by systematic barcoding.

DNA-, RNA-, and peptide-based LNP barcoding systems for tracking multiple LNPs in a single experiment tackle certain aspects of the above-mentioned challenges[17–23]. However, current barcoding strategies have important limitations for high-resolution LNP tracking studies. Although less expensive, DNA barcodes are carried on a separate molecule that is chemically and structurally distinct from mRNA, raising the potential for decoupled distribution of barcodes and cargo, and may give rise to altered LNP properties[18,19]. It was shown that RNA barcodes added to luciferase mRNA correlated to functional delivery with higher accuracy as compared to DNA barcodes[18]. However, luminescence readouts are not suited for high-resolution techniques such as cell sorting and microscopy. mRNA-encoded peptide barcodes have enabled direct quantification of LNP-mediated protein production using mass spectrometry[19]. However, while nucleic acid barcodes can be amplified using ubiquitous techniques like polymerase chain reaction (PCR), peptide barcode abundance cannot be amplified. Therefore, reading the barcodes via mass spectrometry requires relatively large quantities of input cells, making this approach less suitable for cell-specific LNP discovery and high content characterization[19]. These challenges could be attenuated by coupling direct visualization of functional mRNA delivery using fluorescent microscopy with fate-tracing using barcoded mRNA sequencing.

Here, we developed an mRNA barcoding approach that enabled multiplexed in vivo LNP tracking in tdTomato (tdTom) reporter mice for optimization and discovery applications. Due to the compatibility with standardized next-generation sequencing (NGS)- and PCR-based workflows, deep assessment of biodistribution and function could be resolved across space and time. We discovered that functional protein activity was associated with rapid organ enrichment in the liver, lung, and spleen of mice treated with SORT LNPs. This allowed us to pinpoint extrahepatic delivery events missed by previous organ-level studies. Multiplexed spatiotemporal profiling of LNPs using barcoded mRNA was performed to streamline systematic kinetic studies, enabling multiplexed characterization of LNP pharmacokinetics in a single mouse that would have required at least 18 mice if performed one-by-one. Fluorescence microscopy also revealed LNP formulations with hepatic zonal bias, which could have utility for disease modeling and therapeutic applications. Ultimately, multiplexed barcoding platforms hold immense promise for LNP discovery because they reveal tissue-dynamic kinetics and cell-type-specific LNP tropism.

## Results

### Barcoded mRNAs retain functional activity and can be quantified across a wide dose range

Our goal was to create nanoparticles containing barcoded mRNAs that would allow multiplexed administration and cell-specific detection in vivo. We chose to add barcodes to Cre recombinase mRNA due to its compatibility with Cre-Lox reporter systems such as the Ai14 mouse strain[24]. These mice carry a Lox-Stop-Lox tdTom transgene that generates a permanent red fluorescent signal in cells that make Cre recombinase protein. Therefore, barcoded Cre mRNAs enable pooled administration that can be detected and quantified by NGS and tdTom

fluorescence allows for visualization and enrichment of functional delivery events. Together, these tools enable multiplexed high-resolution characterization of LNP biodistribution and functional activity (Fig. 1a).

To create barcoded Cre recombinase mRNAs, we inserted unique 10-nucleotide barcodes and a 20-nucleotide PCR adapter sequence between the stop codon and 3′ untranslated region of a Cre recombinase in vitro transcription (IVT) template DNA (Fig. 1b). The adapter sequence enabled unbiased PCR amplification of the barcoded region, while the barcodes contained a unique identifying sequence that can be readout using qPCR and/or next-generation sequencing. To ensure efficient translation in vivo, mRNAs were synthesized with a template encoded polyA tail, 100% $N^1$-methyl pseudo-uridine substitution, and included a m7G CAP1 structure. After IVT, the size and integrity of barcoded mRNAs were examined using an Agilent Tapestation, which yielded single bands at the expected molecular weight (Supplementary Fig. 1a, b).

We examined whether the barcodes and adapter sequences might influence the functional activity of the Cre mRNA. To study this in vivo, we delivered barcoded or non-barcoded Cre mRNAs to Ai14 mice using liver-targeting LNPs[11] (Supplementary Table 1). We chose to utilize vortex mixing because this method is compatible with high throughput, automatable workflows, and produces high-quality LNPs with comparable in vivo properties to microfluidic mixing[8]. Dynamic light scattering was used to measure the size and dispersity of the barcoded LNPs (Supplementary Fig. 1c). Three days after LNP administration, the Cre-mediated DNA editing efficiency and tdTom expression was quantified using ex vivo imaging and flow cytometry. We found nominal differences in efficiency between the barcoded and non-barcoded Cre mRNAs (Supplementary Fig. 1d, e).

Base modifications such as the $N^1$-methyl pseudo uridine ($\Psi^1$) used here are required for efficient translation of LNP-delivered mRNAs in vivo[25–27], but their effects on standard tissue mRNA quantification techniques are not well established. To ensure that PCR-based detection of $\Psi^1$ IVT mRNA is a reliable and linear quantification method, we generated a standard curve by spiking a known mass of $\Psi^1$ IVT mRNA into total RNA isolated from mouse liver prior to performing reverse transcription, bypassing variables such as LNP delivery and RNA purification. The mass of spiked IVT mRNA spanned five orders of magnitude, from $10^{-5}$ to $10^0$ w/w of total RNA. We then performed qPCR using primers that anneal to the PCR adapter site and ~100 bp upstream of the stop codon. Linear regression analysis found near-perfect correlation between the mass of IVT mRNA in the RT reaction and the resulting fold changes (Supplementary Fig. 2a). To confirm that mRNA LNP delivery could be quantified by extracting RNA directly from target tissues, we intravenously administered liver-targeting LNPs at doses ranging from 0.0001 to 1 mg/kg and collected major organs after 24 h. The qPCR signal in the liver shared a strong correlation with administered doses in vivo, $R^2 = 0.986$ (Supplementary Fig. 2b). Additionally, the tdTom response in the liver was also well correlated with the administered dose, $R^2 = 0.974$ (Supplementary Fig. 2c).

LNP delivery of mRNA involves numerous steps, including tissue distribution, cellular uptake, endosomal escape, and mRNA translation. Prior studies that used DNA- and RNA-based barcodes selected downstream timepoints associated with blood clearance and protein expression[18,28,29], events that could occur minutes to hours after important upstream events take place. Since both DNA and mRNA-based barcode readouts rely on the assumption that biodistribution patterns correlate with functional tropism, we sought to systematically optimize key experimental parameters, including dose and tissue collection time. This approach enabled us to interrogate the relationship between biodistribution and functional activity across a range of conditions. We performed detailed assessments spanning multiple doses and previously uncharacterized early and late timepoints. We utilized SORT LNPs for this goal because the distinct organ delivery

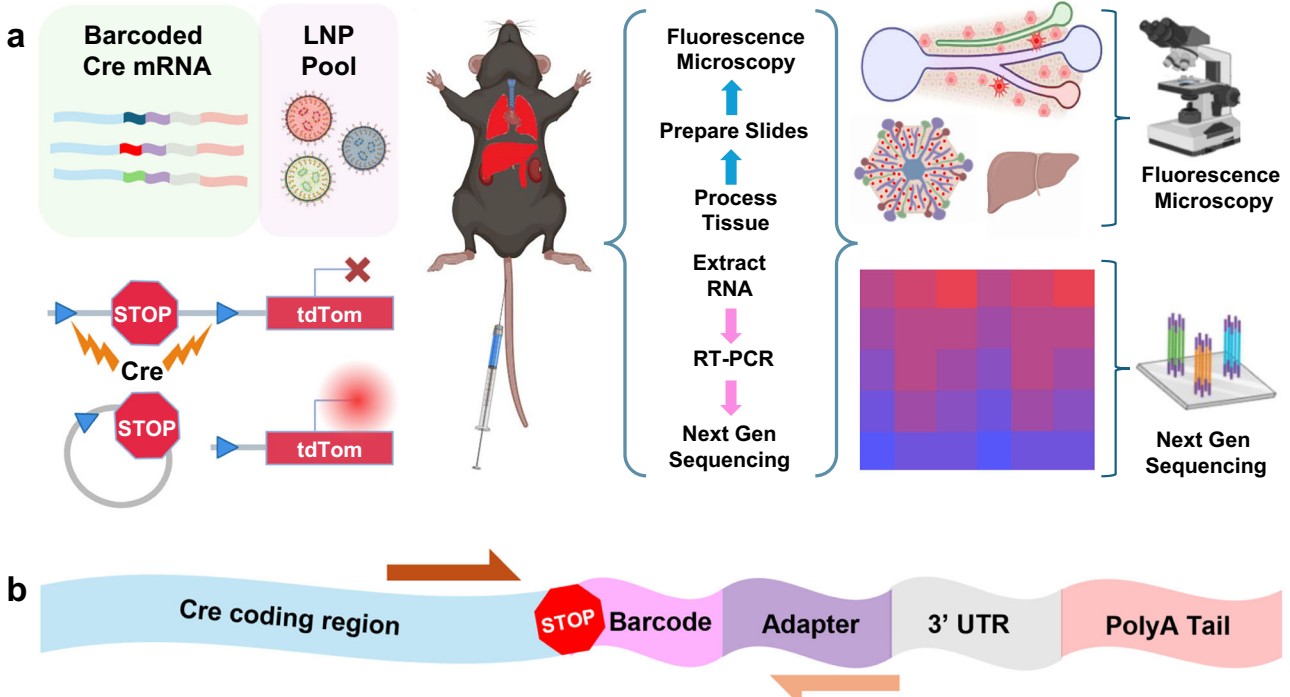

**Fig. 1 | A versatile mRNA barcode platform and workflow was established and validated. a** LNPs are formulated with a unique barcoded Cre mRNA, pooled and administered in multiplex, then the barcodes are recovered in total RNA isolated from cells of interest, reverse transcribed into cDNA, amplified using targeted PCR, and read via next-generation sequencing. Ai14 mice carry a tdTom reporter gene controlled by an upstream LoxP-flanked stop cassette. Cre-mediated recombination of the LoxP sites results in a ~900 bp deletion that enables stable reporter gene expression in cells that make Cre protein. Combining barcoded Cre mRNA with Ai14 mice enables multiplexed characterization of LNP pools. **b** Barcodes were inserted into functional Cre mRNA downstream of the protein coding region with a PCR adapter site for unbiased barcode amplification; arrows approximate the target amplicon.

profiles of these formulations provided a controlled framework to investigate factors such as detection thresholds, saturation dynamics, and the extent to which mRNA biodistribution correlates with downstream protein expression.

Barcoded Cre mRNAs packaged in Lung, Liver, and Spleen SORT LNPs were individually administered IV to Ai14 mice at mRNA doses between 0.001–1.0 mg/kg. Cre mRNA biodistribution was subsequently examined in the lung, liver, and spleen at 1- and 24-h using qPCR. Because successful delivery of Cre mRNA leads to activation of tdTom protein, fluorescence microscopy combined with qPCR quantification can correlate mRNA biodistribution that is nonfunctional (no tdTom expression) and functional (tdTom expression). The tdTom+ area fraction of fluorescent micrographs in regions defined by DAPI signal was used to quantify the tdTom response, a measurement that assumes approximately linear scaling with the number of tdTom+ cells.

Liver SORT LNPs drove efficient, dose-dependent tdTom expression in the liver, with minimal signal in the spleen across the dose range. Some tdTom expression was observed in splenic red pulp at 1.0 mg/kg, possibly indicating hepatic saturation at this dose. No tdTom signal was detected in the lungs at any dose (Fig. 2a and Supplementary Fig. 3a). Cre mRNA levels measured by qPCR showed strong dose dependence at both 1 and 24 h, but were ~100-fold higher at 1 h across the dose range (Fig. 2b, c). Liver tdTom expression correlated with liver Cre mRNA levels at both timepoints; however, the correlation was modestly stronger at 1 h (Fig. 2d, e). Moreover, the distribution of Cre mRNA across the liver, lung, and spleen shared more resemblance to the tdTom distribution when measured at 1 h (primarily liver) than at 24 h (primarily spleen) (Supplementary Fig. 3j). These results demonstrated that the biodistribution of liver SORT LNPs measured 1 h post-injection were a better match for the observed

functional activity in the liver, while the biodistribution at 24 h showed potentially misleading splenic tropism.

Consistent with previous reports, Lung SORT LNPs mediated efficient tdTom expression in the lung across the dose range but also showed some tdTom signal in the liver and spleen at higher doses. (Fig. 2f and Supplementary Fig. 3b). Cre mRNA levels measured by qPCR showed strong dose dependence at both 1 h and 24 h, but were ~100-fold higher at 1 h (Fig. 2g, h). Lung tdTom signal correlated with lung Cre mRNA levels at both timepoints; however, correlation was modestly stronger at 1 h (Fig. 2i, j). While the overall distribution of Cre mRNA resembled the tdTom distribution at both timepoints, more liver and spleen signal were measured by qPCR at 24 h than at 1 h (Supplementary Fig. 3k). These results demonstrated that the observed functional activity of Lung SORT LNPs were reflected in their biodistribution, and suggest that 1 h measurements enable increased signal intensity and modestly improved correlation to protein production versus later measurements.

Spleen SORT LNPs mediated efficient and specific tdTom expression in the spleen, with very few tdTom-positive cells in the liver, and no detectable signal in the lung. The tdTom signal was concentrated in the marginal zones of the spleen, a distinctive distribution compared to Liver SORT or Lung SORT (Fig. 2k and Supplementary Fig. 3c). Cre mRNA levels measured by qPCR showed strong dose dependence and were ~10-fold higher at 1 h than at 24 h (Fig. 2l, m). Spleen tdTom signal correlated with spleen Cre mRNA levels at both timepoints; however, correlation was modestly stronger at 1 h (Fig. 2n, o). The overall distribution of Cre mRNA was similar at both timepoints, showing primarily spleen localization (Supplementary Fig. 3l). Taken together with Liver SORT and Lung SORT, these experiments demonstrated on-target enrichment of Cre mRNA in target organs that improved when measured at 1-h post-injection versus 24 h.

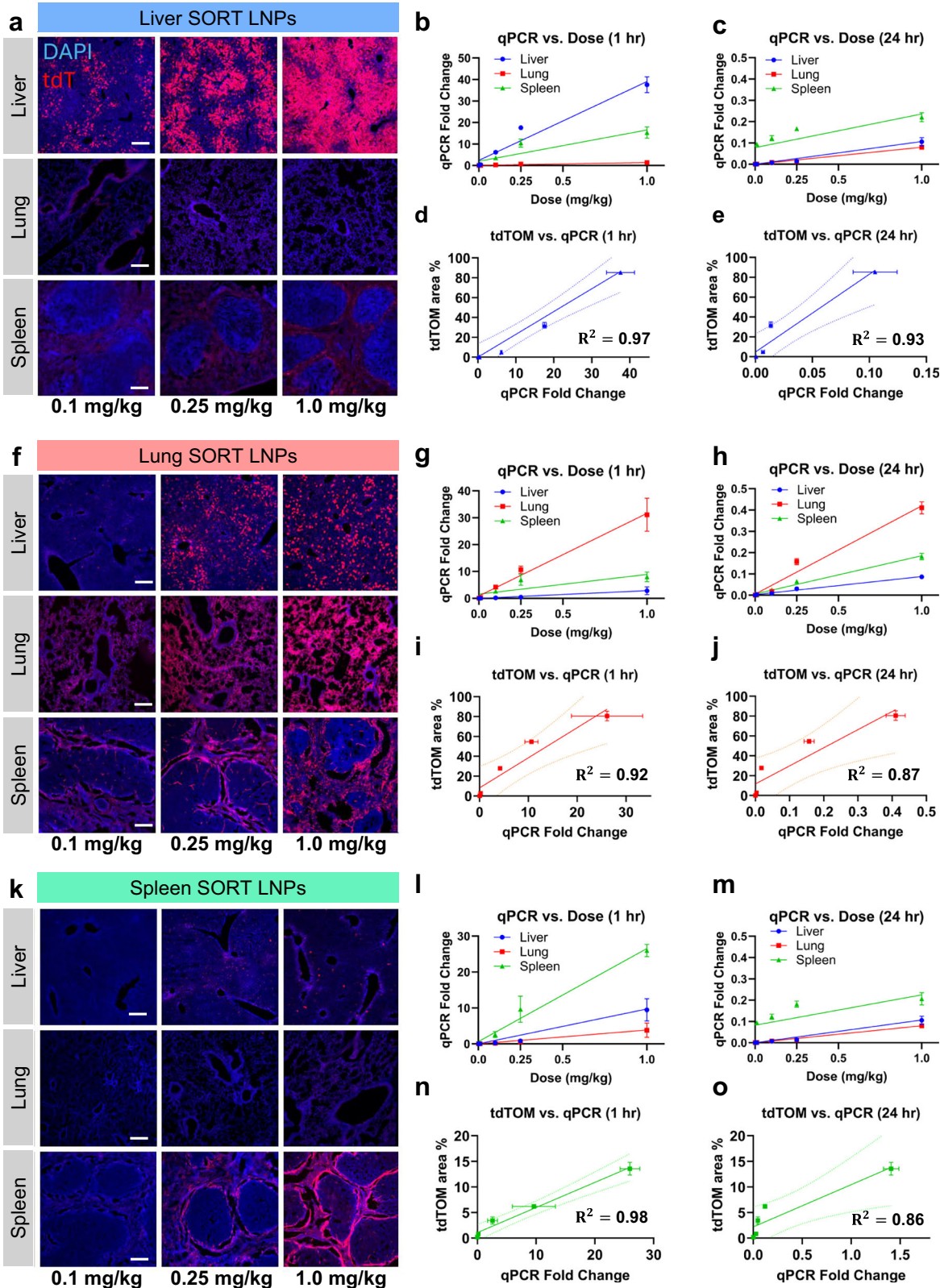

To explore whether these findings could impact a multiplexed barcoding experiment, we designed a test case in which Liver, Lung, and Spleen SORT LNPs were formulated with distinct barcodes and administered as a pool to a single mouse at 0.1 mg/kg per formulation. We harvested the liver, lungs, and spleen at both 1 and 24 h after injection and sequenced barcodes from bulk tissues. Samples harvested at 1 h showed substantial on-target enrichment, and sequencing results correctly ranked the observed functional activity of all three formulations in each organ. In contrast, the on-target enrichment at 24 h was weaker (Supplementary Fig. 3m, n). This study demonstrated that multiplexed barcoding can prioritize formulations with functional activity in target organs and showed that

**Fig. 2 | LNP-mRNA biodistribution correlates with functional activity.**
**a** Endogenous Rosa-tdTom fluorescent micrographs of Ai14 mouse liver, lung, and spleen 24 h post-intravenous (IV) administration of Liver SORT LNPs carrying 0.1, 0.25, or 1.0 mg/kg Cre recombinase shows liver-specific protein production.
**b**, **c** Cre qPCR performed on liver, lung, and spleen 1-h (**b**) or 24-h (**c**) after administration of Liver SORT LNPs at various doses. **d**, **e** Linear regression analysis comparing tdTom signal vs. 1-h qPCR (**d**) or 24-h qPCR (**e**) shows linear correlation that is modestly stronger at 1-h. **f** Endogenous Rosa-tdTom fluorescent micrographs of Ai14 mouse liver, lung, and spleen 24 h post-intravenous (IV) administration of Lung SORT LNPs carrying 0.1, 0.25, or 1.0 mg/kg Cre recombinase.
**g**, **h** Cre qPCR performed on liver, lung, and spleen 1-h (**g**) or 24-h (**h**) after administration of Lung SORT LNPs at various doses. **i**, **j** Linear regression analysis comparing tdTom signal vs. 1-h qPCR (**i**) or 24-h qPCR (**j**) shows linear correlation that is modestly stronger at 1-h. **k** Endogenous Rosa-tdTom fluorescent micrographs of Ai14 mouse liver, lung, and spleen 24 h post-intravenous (IV) administration of Spleen SORT LNPs carrying 0.1, 0.25, or 1.0 mg/kg Cre recombinase shows spleen-specific protein production. **l**, **m** Cre qPCR performed on liver, lung, and spleen 1-h (**l**) or 24-h (**m**) after administration of Liver SORT LNPs at various doses. **n**, **o** Linear regression analysis comparing tdTom signal vs. 1-hour qPCR (**n**) or 24-h qPCR (**o**) shows linear correlation that is somewhat stronger at 1-h. qPCR data (**b–e**, **g–j**, **l–o**) are presented as the mean ± SEM of $N = 3$ mice per dose, per timepoint, per formulation; three wells per measurement. For tdTom quantification: mean ± SEM, $N = 3$ image fields. Correlation analyses (**d**, **e**, **i**, **j**, **n**, **o**) show Pearson's $R$-squared with 95% CI calculated after confirming homoscedasticity. Scale bars (**a**, **f**, **k**): 200 μm.

samples harvested at 1-h post-injection provided a more optimized readout.

## Multiplexed barcoded PK study revealed rapid accumulation and dynamic distribution

Having identified LNP kinetics as a variable that influenced the relationship between mRNA biodistribution and protein production, we next sought to more precisely describe (1) tdTom activation and (2) mRNA kinetics utilizing the mRNA barcode platform. To identify relevant tissue collection timepoints for tdTom expression, we established a timeline of reporter expression following LNP injections. We delivered 0.1 mg/kg barcoded Cre mRNA to Ai14 tdTom mice using either Liver, Lung, or Spleen SORT LNPs. We collected the lung, liver, and spleen from 10 min to 72 h post injection. Although scattered cells in target tissues were tdTom positive 12 h after dosing, robust signal became detectable after 24 h that continued to expand between 24 and 72 h post-injection (Supplementary Fig. 4a–c). We quantified Cre mRNA levels in total RNA extracts of the lung, liver, and spleen at each timepoint using qPCR. Peak mRNA levels were observed at 10 min across tissues, which was the earliest collection timepoint, and diminished substantially at each subsequent timepoint. Although detectable out to 72 h, the majority of transcript abundance was observed within the initial 3 h (Supplementary Fig. 4d–f).

Given the importance of unexpectedly fast accumulation kinetics, we envisioned that this type of LNP optimization experiment would be uniquely suited for mRNA barcoding. We sought to determine whether the kinetics and enrichment of SORT LNPs are differential and hierarchical with respect to functional activity. Because our preliminary kinetics analysis identified the first 3 h post-injection as a pivotal window for distribution and elimination, we chose to focus on this interval for more detailed analysis. A typical experiment to quantify the kinetics of three LNP formulations at 6 timepoints would require at least 18 mice. To overcome this hurdle, we formulated SORT LNPs with distinct barcoded Cre mRNAs in six pools, then administered the pooled LNPs into a single mouse at six timepoints over 3 h so that each barcode encoded the formulation chemistry and the injection time, thus completing an experiment in one animal instead of 18 (Fig. 3a). NGS provides the most accurate and sensitive method to quantify the relative barcode distribution within each sample and was used here for intra-organ barcode quantification. However, NGS cannot be used to compare the barcode distribution between organs because there is no reliable method to normalize sequencing reads between different samples. Therefore, we validated that qPCR can also be used to accurately quantify the barcodes and used this method for quantification of inter-organ biodistribution by normalizing to actin (Supplementary Fig. 5a).

We observed a reduction of circulating Lung SORT barcodes in the blood just 30 s after injection versus liver SORT and spleen SORT (Fig. 3b). Lung SORT barcodes encoding the 30 s timepoint were the most abundantly detected in the lung, indicating rapid lung association (Fig. 3c). Lung SORT blood clearance occurred more slowly

thereafter, which could have been due to reduced liver clearance (Fig. 3d). Lung SORT was enriched versus other LNPs in the lung and heart from 30 s to 3 h and in the kidney from 5 min to 3 h (Fig. 3c, f, g). Spleen SORT blood clearance temporally coincided with accumulation in the liver and spleen, with Tmax at 5 and 10 min, respectively, and splenic enrichment from 5 to 30 min (Fig. 3b, e). Similarly, Liver SORT LNP blood clearance temporally coincided with rapid liver accumulation, reaching Tmax by 10 min, and Liver SORT LNPs remained enriched in the liver from 10 min to 3 h (Fig. 3b, d). The fast mRNA kinetics we observed show that LNPs can rapidly accumulate in tissues where they have functional activity.

qPCR analysis using barcode-specific primers (Supplementary Fig. 5a) revealed distinct spatiotemporal distributions for each SORT LNP. Liver SORT barcodes were initially enriched in the liver but became more abundant in the spleen by 3 h (Fig. 3h). Given that Liver SORT LNPs did not efficiently induce tdTom expression in the spleen, splenic accumulation could indicate an immune clearance mechanism[30]. Lung SORT barcodes remained predominantly in the lung, with minor splenic presence (Fig. 3i). Spleen SORT barcodes accumulated early in both liver and spleen, with more sustained signal in the spleen, leading to increased splenic proportion over time (Fig. 3j). These findings highlight the dynamic and formulation-dependent biodistribution of SORT LNPs. Area under the curve (AUC) calculations showed that, by integrating over time, the measured biodistributions resemble earlier tdTom results (Fig. 3k).

To verify that multiplexed kinetics results are comparable to single-injection experiments, we performed a 72-h time course using the barcoded kinetics approach to quantify Cre mRNA levels in the liver over time. We compared the results to single injections given at the same-timepoints and found that the resulting curve shapes were similar (Supplementary Fig. 5b). We also asked whether multiple injections containing pooled LNPs appreciably altered LNP biodistributions. We compared biodistributions obtained in the context of multiple pooled injections side-by-side with the same-dose, same-timepoint single injections (Supplementary Fig. 5c). We found that the barcode distributions from multiple pooled injections showed similar trends as single-injection experiments. We note that the dose and injection volumes were minimized in these experiments to mitigate potential effects from multiple pooled injections.

To further understand the relationship between biodistribution at early timepoints and functional mRNA delivery, we conducted a follow-up multiplexed pooled kinetics experiment in which a single mouse received multiple injections containing time-coded barcodes delivered by liver SORT, lung SORT, and spleen SORT LNPs. In addition to isolating barcodes in total RNA from bulk tissue, we also profiled ribosome-associated barcodes in the liver, lung, and spleen by adapting and employing a ribosome immunoprecipitation method[31]. The barcode kinetics in total RNA from these tissues were similar to the preceding experiments, with a rapid accumulation phase followed by subsequent elimination. However, the onset of ribosome-associated barcodes was delayed relative to total RNA, most dramatically in the

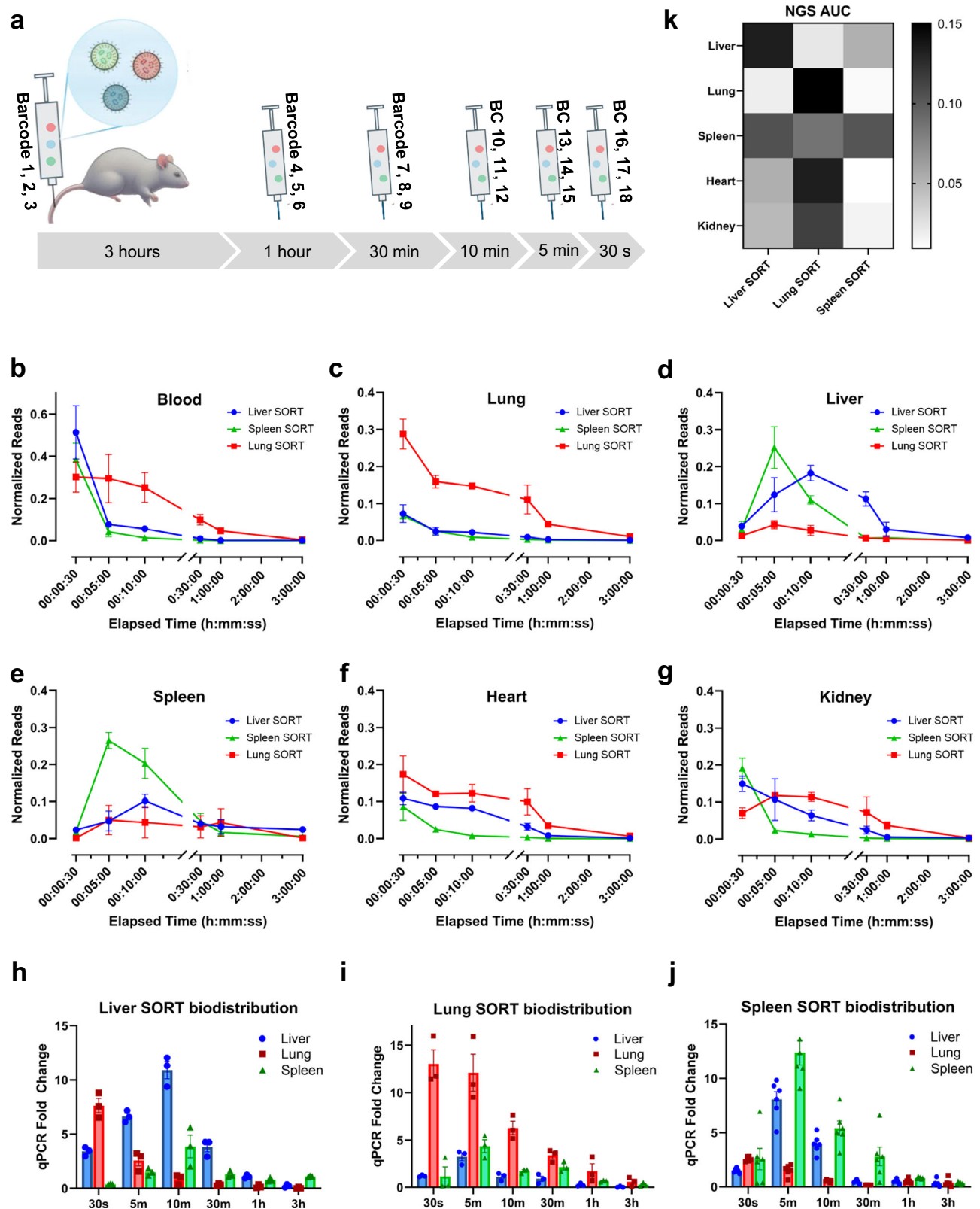

lung, where despite immediate signal peak, ribosome-associated barcodes peaked at 30 min post injection (Supplementary Fig. 5d–f). These results show that the rapid accumulation and enrichment of SORT LNPs in target organs is associated with functional delivery events, and that this functional activity can peak quickly.

To validate qPCR-based quantification of LNP-mRNA biodistribution, we performed fluorescence microscopy using Cy5-labeled mRNA loaded in Liver and Lung SORT LNPs. At both 30 s and 5 min post-injection, we observed distinct intracellular puncta consistent with endosomal localization, indicating rapid cellular uptake (Supplementary Fig. 5g, h). Confocal microscopy revealed Cy5 signal that co-localized with eIF4E, a cytosolic translation initiation factor that interacts with the 5-prime cap of mRNA, in liver samples fixed 20 min after Liver SORT LNP administration (Supplementary Fig. 5i). These

**Fig. 3 | Barcoded pharmacokinetic analysis reveals rapid accumulation kinetics, dynamic distribution, and functional enrichment. a** Lung SORT, liver SORT, and Spleen SORT NPs were formulated with unique barcodes, arranged into six pools, and administered intravenously at six timepoints over 3 h to a single mouse such that each barcode was unique to a formulation and the timepoint it was given. Barcoded pools were given in scrambled order in replicates to reduce any potential barcode bias. **b–g** Normalized NGS reads for each barcode isolated from tissue samples and plotted according to the associated formulation-timepoint. **b** The blood depletion of each formulation temporally coincides with accumulation in target organs. **c** Lung SORT enrichment over liver SORT and Spleen SORT LNPs is maintained in the lung throughout the study. **d** Rapid liver uptake of Spleen SORT and Liver SORT LNPs is followed by fast hepatic clearance of Spleen SORT and

sustained Liver SORT enrichment until 3 hours post injection. **e** Spleen SORT LNPs reach peak splenic accumulation by 5 minutes post-injection and maintain enrichment over Lung SORT and Liver SORT for at least 30 minutes. **f-g** Lung SORT barcodes were enriched vs. Liver SORT and Spleen SORT in **f** kidney and **g** heart tissue. **h–j** Barcoded qPCR was used to map the distribution of SORT LNPs at each timepoint, showing **h** hepatic uptake and splenic migration of Liver SORT LNPs, **i** immediate lung enrichment and gradual splenic migration of Lung SORT LNPs, and **j** rapid Spleen SORT hepatic uptake and clearance followed by splenic accumulation. **k** Area under the curve analysis shows target organ specificity of Liver SORT and Spleen SORT formulations and the relative durability of Lung SORT signal. All data points; $n = 3$ mice; mean ± SEM. Normalized NGS sequencing reads were calculated as follows: (Barcode$_n$ sample reads)/(total sample reads).

findings confirm that mRNA delivery via LNPs initiates rapid tissue-level biodistribution in vivo, supporting the interpretation of early qPCR signals as biologically meaningful.

The use of barcodes for pharmacological profiling enabled us to compare and capture the rapid accumulation kinetics of multiple LNP chemistries head-to-head and highlighted uptake kinetics as a key parameter to optimizing LNP delivery. Furthermore, this approach solved technical challenges such as the requirement for an internal standard to normalize between timepoints, as the barcodes encode the time-axis. This is particularly important for kinetic studies in tissues like blood, or fractionated samples like ribosomes, which lack an obvious internal standard for normalization. Further, we were able to conduct this study using far fewer animals than would normally be required.

## Barcoding streamlines formulation optimization

The ability to distinguish the biological fates between different LNP chemistries is critical to an optimized discovery process. We next asked whether our mRNA barcoding workflow could identify differences between similar LNP formulations. Previously, our lab observed that by titrating the molar fraction of the permanently cationic lipid DOTAP into LNPs, the resulting functional LNP distribution could be gradually shifted from the liver to the lung as a function of DOTAP inclusion percentage[11]. Mechanistically, this phenomenon is at least partially explained by an endogenous targeting mechanism, whereby DOTAP LNPs adsorb Vitronectin from plasma to subsequently internalize in αVβ3 integrin (Vitronectin receptor, Vtn-R) expressing cells that are enriched in the lung[15,16,32]. We decided to use this systematic set of characterized LNPs to test whether our barcoding platform could recapitulate these findings and potentially reveal new insights. To accomplish this, we formulated Liver SORT and DOTAP-containing LNPs with unique barcodes, then administered the pooled series to reporter mice. We collected major organs 1 h post injection and processed samples according to our established barcoding workflow. Separately, we administered each formulation to mice individually with unbarcoded Cre mRNA, then harvested tissues after 24 h to compare with the barcoded data.

In the mice that received a single LNP formulation, we observed the general redistribution of activity from the liver to the lung as a function of DOTAP inclusion percentage, as previously reported[11]. Notably, 30% DOTAP LNPs appeared less efficient than either 20 or 40% DOTAP LNPs in the liver. In addition, we found tdTom+ cell populations in the kidneys of mice treated with DOTAP LNPs, a finding with implications for expanded organ targeting. Recent profiling of αVβ3 integrin (Vitronectin receptor, Vtn-R) expression throughout mouse tissues found that populations of cells within the kidney express Vtn-R[15,16,32,33]. While all of the formulations showed some activity in the spleen, the liver SORT signal was limited to the red pulp. Overall, DOTAP LNPs drove stronger splenic tdTom expression, and the distribution of tdTom extended into the marginal and follicular zones (Fig. 4a).

In our multiplexed analysis, mRNA barcodes also showed a general trend of redistribution from the liver to the lung as a function of DOTAP percentage. Additionally, 30% DOTAP LNP-mRNA barcodes were less abundant in the liver than 20 or 40% DOTAP LNPs. Further, DOTAP LNP barcodes were also enriched versus Liver SORT in the kidney (Fig. 4b and Supplementary Table 2). We plotted the normalized distribution of tdTom signal against the normalized barcode counts and found a strong linear correlation (Fig. 4c). We next asked whether isolating cell populations based on their functional status as tdTom positive or negative would enable further differentiation between formulation activity at the barcode level. To activate the tdTom reporter, we administered the pooled LNP series containing Liver SORT and DOTAP LNPs to Ai14 mice and allowed 24 hours for reporter expression. Prior to harvesting the samples, a second dose of the same LNP pool was given to ensure barcodes were collected during the optimized kinetic window. We then used flow cytometry to collect tdTom-positive and negative cells in the lung, spleen, and kidney (Fig. 4d). We found that the set of barcodes enriched in the tdTom+ populations were associated with enriched functional activity, while barcodes with reduced tdTom activity were enriched in the tdTom-negative populations (Fig. 4e). These multiplex results, performed in a single mouse, recapitulate tdTom reporter results that required 6 mice. Moreover, the detection of kidney tropism associated with DOTAP LNPs demonstrates that our platform can efficiently detect uncharacterized activity within small populations of cells beyond the liver and lung.

To show that these findings are applicable to commercially available LNP formulations currently in clinical use, we examined whether Cre mRNA barcodes could distinguish the functional activity of LNPs based on ALC-0315, SM-102, or LP01 ionizable lipids. To accomplish this, we formulated these LNPs using the clinically described molar ratios of ionizable and helper lipids to carry firefly luciferase mRNA or barcoded Cre mRNA. LNPs were administered IV at 0.1 mg/kg luciferase or 0.01 mg/kg of barcode. We quantified total radiance in ex vivo livers and found that ALC-0315 LNPs resulted in the most luciferase protein production, followed by LP01 and SM-102-based LNPs. We pooled the formulations carrying barcoded Cre mRNAs and administered the pool to Ai14 tdTom reporter mice. The barcode readouts from livers collected 1-h after multiplexed LNP administration in all three pooled mice agreed with the luminescence results obtained using individual mice (Supplementary Fig. 6a–c).

## SORT LNPs differentially distribute within liver metabolic zones
The liver is organized into repeating, roughly hexagonal units termed lobules that consist of a central vein flanked by portal veins. Directionally, blood moves from the portal to the central veins, establishing nutrient and oxygen gradients. Both the gene expression and metabolic function of hepatocytes along this axis are spatially distinct, a phenomenon called liver zonation. Three metabolic zones are well recognized: periportal zone 1, midlobular zone 2, and pericentral zone

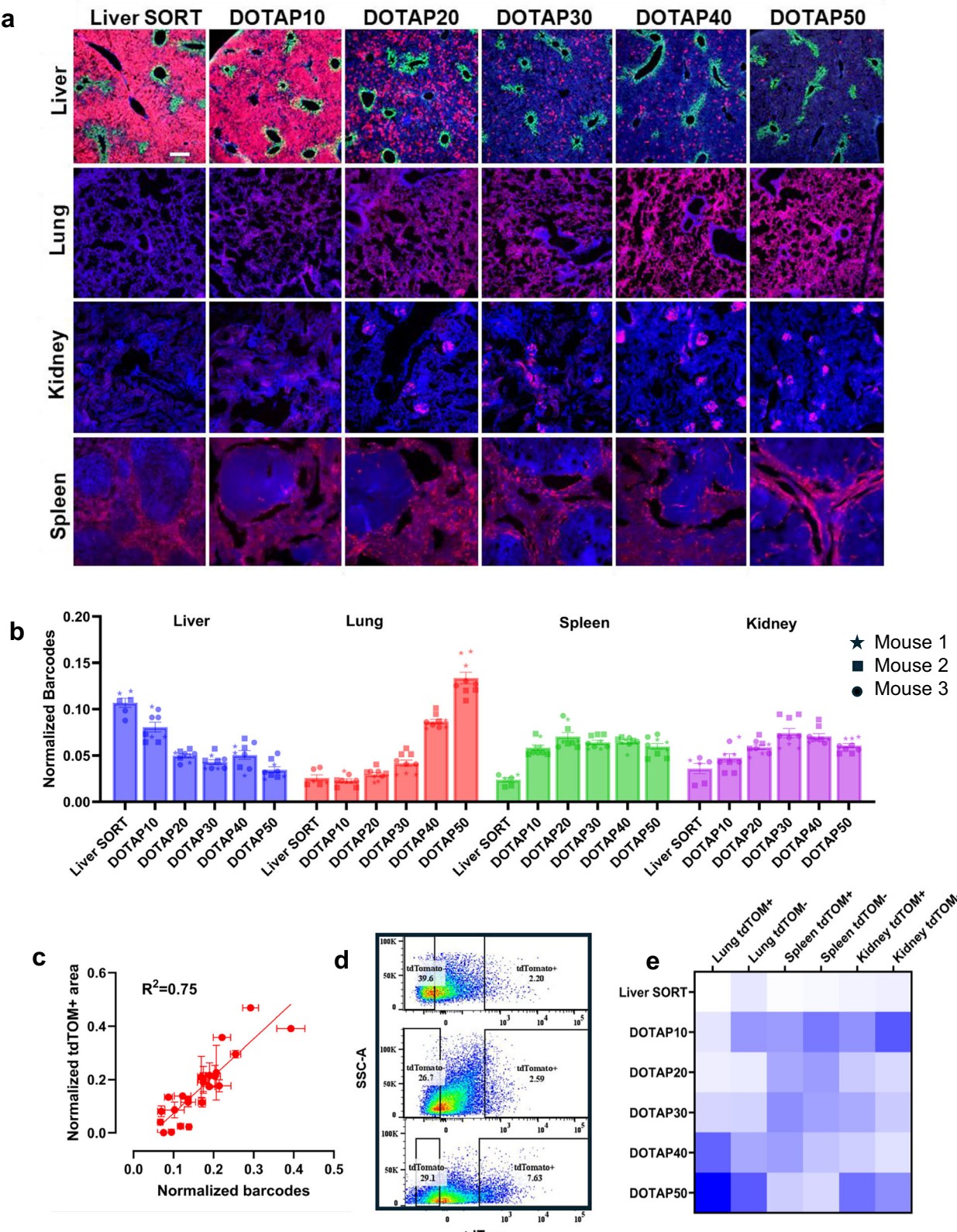

3, each with distinct functions and associated pathologies. The extreme pericentral region expresses the marker gene Glutamine Synthetase (GS), while E-cadherin (Ecad) expression forms a gradient along the portal (highest) to central (lowest) axis[34] (Fig. 5a).

Interestingly, in a new insight, we observed that the activity of Liver SORT LNPs was biased toward periportal and midlobular zones, and excluded from pericentral zones, as tdTom reporter signal showed

limited overlap with the pericentral hepatocyte marker gene (GS). In contrast, LNPs containing 10% DOTAP retained periportal activity, but also had expanded distribution that overlapped with GS+ pericentral hepatocytes. Increasing DOTAP beyond 10 percent de-targeted the liver altogether (Fig. 5b, c). Line scans performed across the central to portal vein axis in a lower dose condition showed that Liver SORT LNPs preferentially drive tdTom expression in zones 1 and 2, with reduced

**Fig. 4 | Barcode distribution faithfully recapitulates delivery profiles of LNPs in a systematic series containing an increasing fraction of SORT lipid and identifies zonal enrichment. a** Endogenous tdTom fluorescent micrographs of the liver, lung, spleen and kidneys of mice administered 0.5 mg/kg Cre mRNA encapsulated in the noted formulations. **b** Multiplexed pooled barcode experiment shows the normalized distribution of barcodes in the liver, lung, spleen, and kidney. LNP pool contained three unique barcodes per DOTAP formulation and two unique barcodes per Liver SORT formulation; N = 3 mice received the pool, mean ± SEM. For statistics, see Supplementary Table 2. **c** Normalized barcode distribution plotted against normalized tdTom signal in the liver, lung, spleen, and kidney shows linear correlation between barcode distribution and tdTom response. Barcode axis: N = 3 mice, mean ± SEM; tdTom axis: Each dot is the normalized mean ± SEM of three image fields per organ per mouse from N = 6 mice. **d** Flow cytometry dot plots with tdTom expression on the x-axis from the lung, spleen, and kidney of mice that received the barcode pool. **e** Heat map showing the normalized barcode distribution in tdTom positive or negative cells isolated using FACS. Barcodes in the tdTom-positive population are enriched for more functionally active formulations vs. tdTom-negative cells. Scale bar (**a**) 200 uM.

zone 3 activity, while the distribution of 10% DOTAP LNPs appears not zonally biased (Fig. 5d).

To recapitulate this finding at the barcode level, we administered a pool of barcoded Liver SORT and DOTAP LNPs, then used flow cytometry to isolate zonated populations based on surface E-cadherin expression[34]. We determined that liver SORT barcodes preferentially distributed to Ecad-high, periportal cells, while 10% DOTAP LNPs were enriched in Ecad-low pericentral cells (Fig. 5e). These liver metabolic zone-specific observations highlight an important consideration for therapeutic applications of LNPs in the liver and could also be leveraged as a tool to study liver phenomena such as metabolic zonation. Moreover, this experiment demonstrates the utility of the barcode platform to distinguish cell-type-specific trends in nanoparticle-mediated mRNA delivery in a multiplexed experiment.

## Discussion

Lipid nanoparticles have emerged at the forefront of nucleic acid delivery platforms. However, the discovery and optimization of LNPs is challenging due to the vast array of input parameters. We sought to aid the discovery process by developing an mRNA barcoding approach that can track multiple LNPs in the same experiment. To capture the biological fate of barcoded LNPs with high resolution, we chose to barcode Cre mRNA due to the suitability of Ai14 tdTom reporter mice to yield a stable functional signal that is amenable to cell sorting and fluorescence microscopy. We demonstrate that this approach provides a framework for multiplexed LNP tracking in defined cell populations, with the capacity to encode spatial and/or temporal information. This substantially expands the utility and potential applications of mRNA barcoding beyond its use as a primary screen.

We performed time-resolved tracing of SORT LNP biodistribution and functional tropism to optimize assay parameters and define the impact of LNP kinetics on functional activity. We discovered surprisingly fast LNP accumulation kinetics, which occurred within seconds to minutes after IV administration. Further, LNP uptake kinetics correlated with functional activity such that downstream protein production was associated with rapid accumulation. Since past barcoding studies have only examined timepoints of 4 h or later post-injection, detection of LNP delivery in tissues may have been missed[29]. This previously unknown misalignment of mRNA LNP kinetics offers a path forward to improve future barcode screening workflows. Indeed, we noted that much of the mRNA signal was lost after ~3 h in this study, demonstrating the fast pharmacodynamics of delivered Cre mRNA. It is worth noting that the specific mRNA accumulation, elimination, and degradation kinetics reported here could be influenced by a variety of factors, such as the base modifications, m7g cap1 structure, untranslated regions, or polyA tail, among others, used in this study. Additionally, further screening must be performed to generalize LNP kinetics to a broader range of chemistries beyond SORT formulations. Studying generalizability to other mRNA modifications and nanoparticle carriers will be an important future goal. These results show that earlier assay timepoints are more optimized for multiplexed LNP barcoding, and these insights enable enhanced predictive power while demonstrating the utility of barcoding to conduct resource-intensive PK/PD studies. Notably, the optimal timepoints we identified here were much earlier than previously reported barcoding studies.

Discovery and formulation engineering campaigns must often distinguish between nanoparticles with similar properties, and we showed that our mRNA barcoding approach can streamline this process. We applied our approach to a series of DOTAP LNPs in the liver, lung, kidney, and found that barcode data recapitulated one-by-one results including subtle differences in the efficacy of mid-series formulations that resulted in reduced functional activity. We showed that this approach can identify LNPs with novel activity when applied to new in vivo targets. Using the barcoded mRNA platform, we identified small populations of cells within the kidney that were successfully transfected by DOTAP LNPs. Fluorescent microscopy revealed that the transfected kidney cell populations were relatively small when compared to the whole liver or lung and could be easily missed using traditional discovery approaches such as ex vivo imaging. Additionally, integration with spatial flow cytometry[34] enabled us to delineate zonated hepatic LNP delivery at the barcode level. Fluorescent microscopy validated these spatial trends, highlighting the power of this approach for comprehensive, cell-level characterization of nanoparticle delivery in vivo. Overall, these findings underscore the versatility of the barcoding platform as both a discovery and validation tool in nanoparticle research.

## Methods

### Barcoded LNP formulation

Barcoded Cre mRNAs were packaged in 5-component SORT lipid nanoparticles using previously described lipids and formulation conditions[8,11]. Briefly, the 5A2-SC8 ionizable amino lipid, DOPE (Avanti Polar Lipids Cat. No. 850725), Cholesterol (Sigma-Aldrich Cat. No. C3045), DMG-PEG-2000 (Avanti Polar Lipids Cat. No. 880151), and a SORT lipid were dissolved in ethanol at previously optimized lipid molar ratios of 15:15:30:3:X respectively, where X is either 15.75 (20%) DODAP (Avanti Polar Lipids, 890850 O), 63 (50%) DOTAP (Avanti, 890890 P), or 15.75 (20%) 18PA (Avanti, 840875 C). The lipid-containing organic phase was rapidly mixed with an aqueous phase containing RNA diluted in 10 mM pH 4 citrate buffer at a 3:1 (aq.:EtOH) ratio using vortex mixing[8] or a NanoAssemblr Benchtop system (Precision Nanosystems) at a 40:1 lipid:nucleic acid mass ratio. LNPs were dialyzed in PBS for at least 2 h prior to use.

### Barcoded mRNA production

Barcodes and a PCR adapter were cloned into an in vitro transcription (IVT) vector encoding Cre recombinase with a DNA-encoded polyA tail using Q5 SDM kit according to the manufacturer's instructions (New England Biolabs, E0554S). Purified barcoded Cre IVT template plasmids were linearized for 4 h via restriction digestion, and mRNAs were transcribed using Megascript SP6 IVT kit (Thermo Fisher, AM1330) with 100% $N^1$-methyl pseudo uridine substituted for uridine. An m7G (CAP1) structure was added to purified mRNA using ScriptCap Cap1 Capping System (CellScript, C-SCCS1710) according to the manufacturer's instructions.

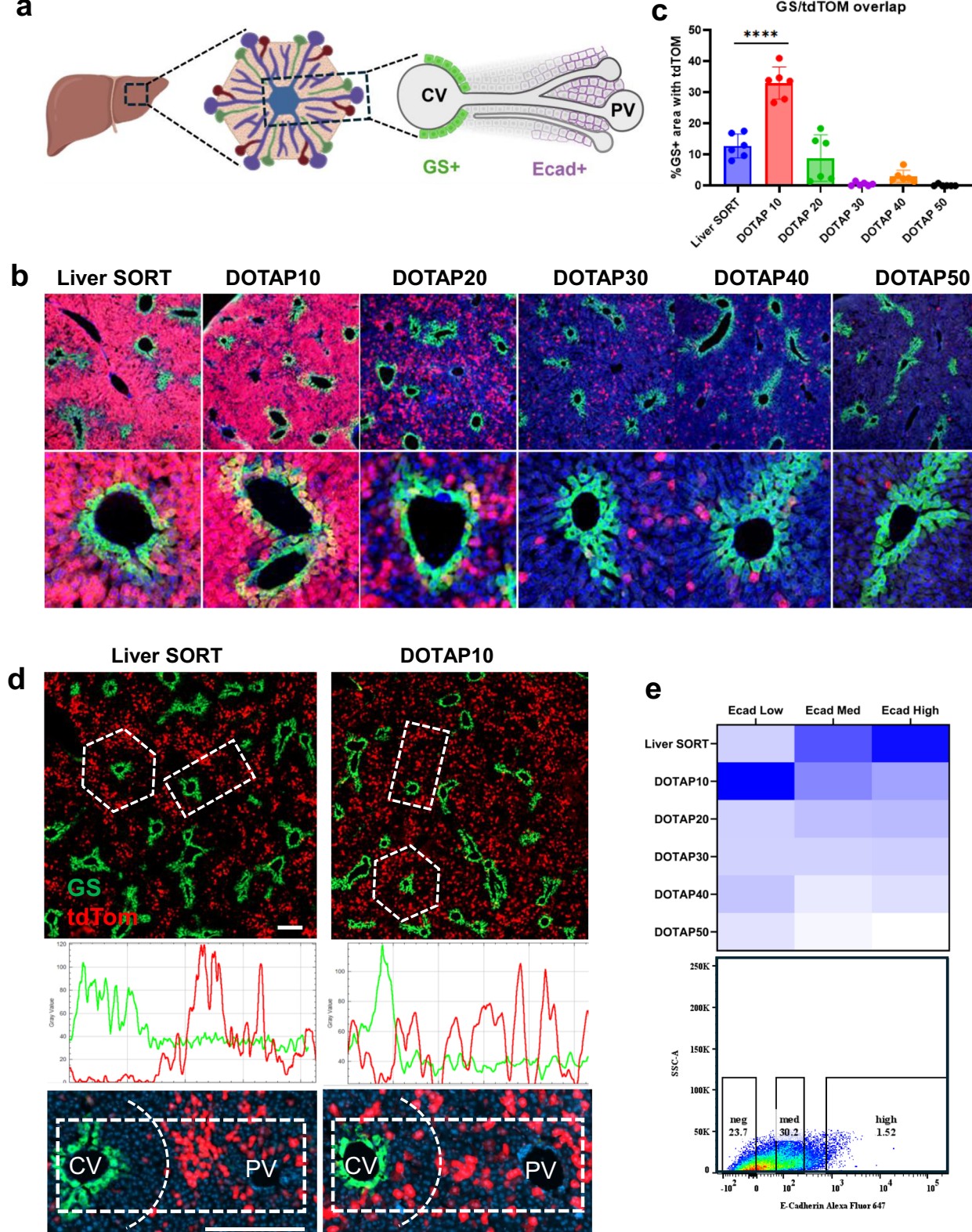

## Flow cytometry

Organs were minced using a sterile blade and homogenized in 250 μL of 1× digestion medium (45 units/μL collagenase I, 25 units/μL DNase I, and 30 units/μL hyaluronidase). The solution was transferred into a 15 mL tube that contained 5–10 mL of 1× digestion medium. Next, the solution containing cells was filtered using a 70 μm filter and washed once with 1× PBS. A cell pellet was obtained by centrifuging for 5 min at a speed of 300×g at 4 °C. The supernatant was removed, and the cell pellet was resuspended in 2 mL of 1× red blood cell lysis buffer (BioLegend, 420301) and incubated on ice for 5 min. After incubation, 4 mL of cell staining buffer (BioLegend) was added to stop red blood cell lysis. The solution was centrifuged again at 300×g for 5 min to obtain a cell pellet. The cells were washed twice with 1 mL 1X PBS, then resuspended in 500 μL 1X PBS for flow cytometry analysis. Ghost Dye Red

**Fig. 5 | Spatial mapping of LNP activity across liver zonation. a** Schematic illustrating the liver architecture at increasing magnification: whole liver, liver lobule with a central vein surrounded by six portal triads, and a central-to-portal axis with glutamine synthetase (GS)-positive cells near the central vein and Ecad-positive cells near the portal tract. **b** Reporter imaging reveals LNP activity across metabolic zones in response to DOTAP titration. Liver SORT transfects primarily periportal regions, sparing GS+ pericentral cells. 10% DOTAP inclusion enhances transfection in GS+ cells, while higher DOTAP levels (20–50%) reduce liver targeting. **c** Quantification of GS and reporter signal colocalization highlights zonal transfection biases of each formulation. $N = 6$ image fields per condition, mean ±

SEM; one-way ANOVA with Tukey's test for multiple comparisons, $p < 0.0001$. **d** Fluorescent line scans over the central-to-portal axis of a liver lobule show no tdTom GS overlap when Liver SORT LNPs are dosed at 0.1 mg/kg and stochastic distribution with GS overlap when 10% DOTAP LNPs are given at the same-dose. Line scan profiles are representative examples of five lobules analyzed per formulation. **e** Heat map and FACS dot plot of multiplexed barcode analysis showing flow cytometric sorting by Ecad expression and barcode enrichment patterns: Liver SORT enriches in Ecad high-expressing cells (periportal); both formulations are represented in Ecad mid-expressing (midlobular) population, and 10% DOTAP LNPs are enriched in the Ecad low (pericentral) expressing cells. Scale bar (**b**, **d**) 200 uM.

---

780 (Tonbo Biosciences, 13-0865-T500) was used to discriminate live cells. Where applicable, the sample was subjected to antibody staining in preparation for flow cytometry. The cells were analyzed using a LSRForessa SORP machine (BD Biosciences).

### RNA extraction

About 30 mg of whole tissue was added directly to 1 mL of ice-cold TRIzol reagent (Thermo Fisher, 15596026) and homogenized in a bead beater homogenizer for 45 s. The homogenate was allowed to lyse on ice for 15 min, then 0.2 mL of chloroform was added and vortexed for 15 s. Phase separation proceeded on ice for 5–10 min, then samples were centrifuged at 12,000×$g$ for 15 min. The upper aqueous phase containing RNA was carefully pipetted into a separate tube, and the RNA was precipitated using one volume of isopropanol, then centrifuged for 15 min at 12,000×$g$. RNA pellets were resuspended in nuclease-free water, and the RNA yield and purity was calculated using UV-Vis spectrophotometry.

### Ribosome pulldown

Ribosome purification proceeded as previously described[31] with modifications to adjust for larger tissue volumes. Organs were digested as described for flow cytometry in the presence of 0.1 mg/mL cycloheximide. Cells were filtered through a 70-uM strainer then lysed in 100 μL lysis buffer (20 mM Tris–HCl, pH = 7.4, 250 mM NaCl, 15 mM MgCl$_2$), 0.1 mg/mL cycloheximide, 0.5% Triton X-100, 1 mM DTT, 0.5 U/μL RNAse inhibitor (Thermo Fisher, AM2696), 0.024 U/μL TURBO DNase (Life Technologies, AM2222), 1 × Protease Inhibitor (Sigma, P8340)). About 2 μL of biotinylated y10b antibody (Thermo Fisher, MA516060) was added to each sample and incubated for 4 h on an inversion rotor at 4 °C. After the incubation, 100 uL of prewashed streptavidin-coated magnetic beads (NEB S1420S) were added to each sample. Beads were washed 3x in 1 mL of wash buffer (20 mM Tris–HCl (pH 7.4), 250 mM NaCl, 15 mM MgCl$_2$, 1 mM DTT, 0.1 mg/mL cycloheximide, and 0.05% v/v Triton X-100). Samples were incubated with beads for 30 min at 4 °C with inversion. Samples were applied to a magnet and washed 3x in 1 mL of wash buffer with resuspension between washes. After the final wash, beads were resuspended in 50 uL of RNAse-free water and heated in a thermocycler at 65 °C for 10 min. The magnetic beads were removed using a magnet, and the supernatant containing ribosome-associated RNA proceeded directly to reverse transcription. Barcodes were quantified using qPCR.

### RT-qPCR

RNA extracts were reverse transcribed into cDNA using the iSCRIPT cDNA synthesis kit (Bio-Rad, 1708891) according to the manufacturer's instructions. cDNA libraries were used as template DNA for quantitative PCR using iTaq supermix (Bio-Rad, 1725124) as follows: for the detection of all barcoded Cre mRNA irrespective of barcode, the forward primer sequence used was:
5′-ACCGGAGATCATGCAAGCTGG-3′
the reverse primer sequence was:
5′-TTGCTAGGACCGGCCTTAAAGC-3′.

The reverse primer sequence annealed to the PCR adapter site and the forward primer sequence annealed -150 nt upstream. Samples were normalized to the following primers targeting Actin:
5′-TGTTACCAACTGGGACGACA-3′ and
5′-ACCAGAGGCATACAGGGACA-3′.

Reactions were run on a CFX 384 thermocycler using the following cycling conditions: 1x [3 min @95 °C], 39x [10 s @95 °C, 30 s @65 °C, plate read]. CT values were calculated using Bio-Rad CFX Manager v3.1 software. 2^- dCT (fold change) values were calculated as follows: 2^-[CT(Cre)-CT(Actin)].

Some barcoding experiments were readout via qPCR using primers that annealed to the barcoded region and compared using the relative dCT values. The forward primer was the same as above; the reverse primers used are listed below. PCR conditions were the same as described above.

| Name | Barcode sequence | Reverse primer sequence |
| --- | --- | --- |
| BC1 | 5′-CTGTTTCTCC | 5′-ACCGGCCTTAAAGCGGAGAAAC |
| BC2 | 5′-AATGGGAGAC | 5′- CCGGCCTTAAAGCGTCTCCC |
| BC3 | 5′-AAGGTGCGTT | 5′-CGGCCTTAAAGCAACGCACCTT |
| BC4 | 5′-GGCAATTGTC | 5′-GGACCGGCCTTAAAGCGACAATTG |
| BC5 | 5′-AAGTACCCTC | 5′-ACCGGCCTTAAAGCGAGGG |
| BC6 | 5′-GGGGTGGTTA | 5′-GCCTTAAAGCTAACCACCCC |
| BC7 | 5′-AAGTGCGTCA | 5′-GGCCTTAAAGCTGACGCACTT |
| BC8 | 5′-ATTTCAAAAC | 5′-GACCGGCCT-TAAAGCGTTTTGAAAT |
| BC9 | 5′-ATCATCGTGG | 5′-GGCCTTAAAGCCCACGATGAT |
| BC10 | 5′-ACCTCCGCAA | 5′-GCCTTAAAGCTTGCGGAGGT |
| BC11 | 5′-GGGTGTCGGG | 5′-CCTTAAAGCCCCGACACCC |
| BC12 | 5′-GTATAATTTG | 5′-TAGGACCGGCCTTAAAGC-CAAATTATAC |
| BC13 | 5′-ATAGCTAGTC | 5′-CCGGCCTTAAAGCGACTAGCTAT |
| BC14 | 5′-ACCGCAGAGG | 5′-CCTTAAAGCCCTCTGCGGT |
| BC15 | 5′-AGCATATGAG | 5′-CCGGCCTTAAAGCCTCATATGCT |
| BC16 | 5′-CGCAGAATAA | 5′-CCGGCCTTAAAGCTTATTCTGCG |
| BC17 | 5′-TGAGGACCAT | 5′-CGGCCTTAAAGCATGGTCCTCA |
| BC18 | 5′-AGCTTCTCGT | 5′-GGCCTTAAAGCACGAGAAGCT |
| BC19 | 5′-ATAGTTGCGG | 5′-GGCCTTAAAGCCCGCAACTAT |
| BC20 | 5′-GCATATGTGA | 5′-CCGGCCTTAAAGCTCACATATGC |
| BC21 | 5′-AACATGGGGT | 5′-GGCCTTAAAGCACCCCATGTT |

## Next-generation sequencing (NGS)

RNA extraction and cDNA synthesis were performed as described above. Full-length Cre amplicons were obtained via PCR with Q5 DNA polymerase (NEB, M0491L) using primers that anneal the 5′ UTR and 3′ UTR, followed by agarose gel electrophoresis and purification of the resulting 1.3 kb DNA band. NGS libraries were prepared using a two-step PCR procedure with the following primer sets to amplify the barcode region and add Illumina flow cell adapters, sequencing adapters, and indexes:

PCR1:

5′-GTCATGAACTATATCCGTAACC-3′

5′− 5′ GTGACTGGAGTTCAGACGTGTGCTCTTCCGATCTTTGCTAG GACCGGCCTTAAAGC-3′

PCR2:

5′-AATGATACGGCGACCACCGAGATCTACACTCTTTCCCTA-CACGACGCTCTTCCGATCTGTCATGAACTATATCCGTAACC-3′

5′-CAAGCAGAAGACGGCATACGAGAT NNNNNNNN GTGACTG-GAGTTCAGACGTGTGC-3′

Where NNNNNNNN is the sample index.

Indexed libraries were pooled and submitted to the McDermott NGS core at UT Southwestern.

## Barcode quantification

Reported barcode values are presented as the proportion of reads within a given sample index normalized to an LNP pooled input sample. Alternatively, some experiments used barcode-specific PCR primers to read the barcodes using RT-qPCR according to the methods described above. For qPCR quantification, the reported barcode values are presented as the log2 fold change versus actin, normalized to an input pool.

## Immunofluorescence microscopy

Tissues were fixed with 10% formalin at 4 °C overnight. The fixed specimens were cryoprotected with a graded series of sucrose and then embedded in tissue-freezing medium. Frozen tissue blocks were cut into 12-µm cryosections. For immunostaining, 10% goat serum in 0.1% Triton X-100 1× PBS solution was used for blocking, and 3% goat serum in 0.1% Triton X-100 1× PBS solution was used for primary/secondary antibody incubations. Sections were incubated in primary antibody at 4 °C overnight, then washed 3 × 10 min in PBS to remove unbound antibody. Samples were incubated in secondary antibody for 1 h at RT, then washed 3 × 10 min in PBS. ProLong Gold Antifade Reagent with DAPI (Thermo Fisher) was used to mount the samples and visualize nuclei. Imaging of the samples was carried out on a Zeiss Z1 slide scanning microscope or a Leica DMi8 inverted microscope. Original images were processed using Zeiss ZenBlue software, Leica LasX software, or exported in tif format and processed using ImageJ.

## Animal models

Animal experiments were approved by the Institutional Animal Care and Use Committee of The University of Texas Southwestern Medical Center under APN 2015-101118-G, APN 2015-100933, APN 2016-101430, and APN 2017-102283 and were consistent with local, state, and federal regulations as applicable. B6.Cg-Gt(ROSA)26Sortm14(CAG-tdTomato) Hze/J (also known as Ai14) mice were obtained from The Jackson Laboratory (007914) and bred to maintain homozygous expression of the Cre reporter allele that has a loxP-flanked STOP cassette preventing transcription of a CAG promoter-driven red fluorescent tdTomato protein. Following Cre-mediated recombination, Ai14 mice express tdTomato fluorescence. All animals were maintained on a 12/12-h light/dark schedule at a mean temperature of 22 °C and fed with ad libitum normal chow. All experiments were performed in at least three biological replicates. Male and female mice aged 6–12 weeks were used for experiments. Age and sex were distributed equally across experimental groups.

Lateral tail vein injections were performed under physical restraint. Sequential tail vein injections for multiplexed kinetics study were performed as follows: (1) Each injection contained 0.01 mg/kg/formulation, 0.03 mg/kg total mRNA dissolved in 50 uL PBS. (2) Injections at 3 h, 1 h, and 30 m timepoints were administered under physical restraint sequentially in one lateral tail vein (distal to proximal), while 10 m, 5 m, and 30 s timepoints were delivered under isoflurane anesthesia in the opposite vein, also distal to proximal. (3) Following the last injection, mice were immediately euthanized by cervical dislocation. (4) Blood collection was immediately performed via cardiac puncture using a 25-G needle inserted into the left ventricle until ~100 uL was drawn. Whole blood was used for downstream analysis by directly adding 1 mL of TRIzol and following the RNA extraction procedure described above. (5) Major organs were immediately collected in the following order: heart, lung, liver, spleen, and kidneys. Utensils were briefly wiped between organs to avoid cross-contamination. No perfusion was used prior to collecting tissues; tissue samples were added directly to TRIzol and processed as described for RNA extraction. (6) Replicate injection times were staggered to facilitate handling only one mouse at a time.

## Reporting summary

Further information on research design is available in the Nature Portfolio Reporting Summary linked to this article.

## Data availability

The sequencing data generated in this study have been deposited in the NCBI Sequence Read Archive (SRA) database under accession code PRJNA1373256. All other data supporting the findings of this study are available within the paper and the supplementary information files.

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

## Acknowledgements

We acknowledge financial support from the National Institutes of Health (NIH) National Institute of Biomedical Imaging and Bioengineering (NIBIB) (R01 5R01EB025192-06), National Cancer Institute (R01CA251982), the Mark Foundation for Cancer Research (21-003-ELA), the Moody Medical Research Institute, UTSW Presidential Scholar Program, and Simmons Comprehensive Cancer Center, and a collaboration agreement with Pfizer, Inc. We also acknowledge support from the UTSW Small Animal Imaging Resource (NCI P30CA142543).

## Author contributions

Conceptualization: D.J.S., H.Z., and S.T.M.; Methodology: D.J.S., H.Z., S.T.M., and X.L.; Investigation; S.T.M, X.L., A.V., Y.S., S.C., and J.S.; Manuscript; D.J.S., H.Z., S.T.M., X.L., L.F., Y.S., and A.V.

## Competing interests

The authors acknowledge competing interests in ReCode Therapeutics (D.J.S.), Signify Bio (D.J.S. and L.F.), Newlimit (H.Z.), Chroma Therapeutics (H.Z.), Quotient Therapeutics (H.Z.), and Jumble Therapeutics (D.J.S. and H.Z.). The remaining authors declare no competing interests.
