## [Transparent Peer Review file · Nature Communications]

Multiplexed lipid nanoparticle barcoding reveals tissue-dynamic kinetic insights and enriched cellular tropism in hepatic zones

Corresponding Author: Professor Daniel Siegwart

Version 0:

Reviewer comments:

Reviewer #1

(Remarks to the Author)

This manuscript by Moore et. al. presents an innovative approach utilizing multiplexed lipid nanoparticle (LNP) mRNA barcoding to enhance the characterization of tissue-specific biodistribution and functional mRNA delivery. By employing Cre recombinase mRNA barcoding and tdTomato reporter mice, the study provides a novel high-throughput method for evaluating LNP pharmacokinetics and cellular tropism across multiple organs. The authors demonstrate that barcoded LNP tracking can streamline kinetic studies, identify distinct biological outcomes, and uncover previously overlooked hepatic zonal bias and extrahepatic tropism. These findings hold substantial implications for optimizing mRNA delivery platforms for therapeutic applications. This study makes a strong contribution to the field of LNP by addressing critical challenges in pharmacokinetics evaluation. The characterization of barcoding mRNA in vivo kinetics also shows great impact on mRNA barcoding field to demonstrate mRNA have the same robustness as DNA barcoding in relatively shorter period. I recommend publication in Nature Communications after consideration of the minor questions below:

Technical Questions:

1. Figure 2: The fluorescence images for the liver, lung, and spleen were presented, yet the correlation between tdTomato (tdTOM) and qPCR was only shown for the targeted organ. To improve the robustness of the analysis, it would be beneficial to include correlation data for the liver, lung, and spleen in each organ. Ideally, for non-targeted organs, a weaker correlation would further validate the specificity of this system.
2. Figure 2: Additional data points are needed (more than n=2) to provide stronger support for the conclusions drawn.
3. Figure 3: There appears to be an inconsistency between the curve plots and the bar graphs. For instance, in Figure 3f, the liver SORT data shows a peak level at approximately 10 minutes, whereas in Figure 3h, the peak level for the spleen is at 3 hours. A similar inconsistency is observed between Figures 3f and 3j. The authors should clarify these discrepancies.
4. Figure 4b and 4c: Additional data points are needed in both figures to provide stronger support for the conclusions drawn.
5. The conclusion that liver SORT LNPs are restricted to periportal hepatocytes while LNPs containing 10% DOTAP expand distribution to pericentral hepatocytes could be better supported. Consider addressing: 1) The second row of images in Figure 4a appears vague. A high-resolution image is necessary for better visualization. 2) Quantification of distribution in pericentral hepatocytes is required. Flow cytometry analysis may be needed to better quantify tdTomato+ GS+ hepatocytes, which would strengthen the conclusion.

Minor Technical Questions:

1. In Figure 2, although the biodistribution of Lung SORT LNPs was lung-centered, there was still a lower correlation of qPCR results to 24h data. Why?
2. Please also include spleen in Figure 4 / supplementary figure as it is a major organ relevant to validating the robustness of the Cre mRNA barcoding system.
3. While all the microscopy images are striking, consider including some quantification of the signal in these images to further support the conclusions made about the kinetics of Cre mRNA signal vs Cre mRNA accumulation.
4. Please include citations for why you think immune cells are trafficking from the liver to the spleen. Has this been observed before?
5. As we now know, protein corona around LNPs is an important determining characteristic in their tropism. With this in mind, why do you think you observe kidney tropism with addition of DOTAP in your LNP?

Statistical Questions:

1. Statistical tests are largely missing from the current studies. Please add statistical analyses to all studies possible to show any significance.

Stylistic Recommendations:

1. Although Figure 1e shows near-perfect correlation, consider moving it to the supplementary information as information about mRNA modifications does not seem to add too much value to your discussion about multiplexed barcoding and validation.
2. Consider rephrasing “Cre mRNA signal was detectable within the first 3 hours” to “Cre mRNA levels in tissue were detectable within 3 hours” as signal implies fluorescence or some other kind of visual signal.
3. In the section “Multiplexed barcoded PK...”, please clarify what “study mice” are.
4. In figure 3, please clarify the x axes as 00:05 is not in hours. Also indicate where the 30 second timepoint is.

Reviewer #2

(Remarks to the Author)

The study by Moore et al. presents an mRNA barcoding strategy for LNP screening at the tissue level. The authors designed an mRNA barcode encoding Cre recombinase, which triggers tdTomato expression in Ai14 mice, allowing for visualization and enrichment of functional delivery events. The barcoded mRNA enables multiplexed LNP tracking, with its distribution quantified using NGS and qPCR. The study examines the biodistribution of Selective Organ Targeting (SORT) LNPs in the liver, lung, and spleen over time. However, many critical issues limit the novelty and scientific impact of this work. First, the mRNA barcoding design closely follows a previously published mRNA barcode in the Journal of Controlled Release (PMID: 31678653), with the only modification being the replacement of the ORF from luciferase to Cre recombinase. This represents only a minor conceptual advance over prior work, as the study does not introduce any new LNP formulations or provide mechanistic insights into LNP design and function. The authors argue that DNA barcodes, being separate from mRNA, could lead to decoupled biodistribution and altered LNP properties, but this hypothesis was not experimentally tested. The claim remains speculative, especially given that prior studies have successfully used DNA barcoding to screen hundreds of LNPs in a single animal, whereas the current mRNA barcoding approach was only demonstrated with three types of LNPs in most of its studies. Notably, the barcoding approach did not uncover any previously unknown mechanisms of LNP delivery. The only new finding (Liver SORT LNPs primarily went to periportal hepatocytes in Fig. 4A) was identified using standard fluorescence imaging in mice individually dosed with single LNP formulations, rather than through multiplexed barcoding. This contradicts the claimed utility of the barcoding approach. The study also does not identify any new LNP formulations with improved properties and instead applies the barcoding technique to pre-existing SORT LNPs. The study primarily confirms already expected biodistribution patterns rather than providing new discoveries about LNP behavior or design principles. While the barcoding approach may be useful for small-scale LNP kinetic studies, its practical advantages over prior barcoding methods remain unproven. Overall, this study provides an incremental methodological refinement rather than a significant conceptual or technical progress. It does not convincingly demonstrate an advantage over existing DNA or peptide barcoding methods. The lack of essential controls and sensitivity validation as well as the failure to confirm cytosolic mRNA release limit the impact of this work.

1. The authors claim that the Cre mRNA barcode design inherently links organ-level biodistribution with functional delivery; however, this is not entirely accurate. If multiple LNPs containing distinct Cre mRNA barcodes enter the same cell and successfully induce tdTomato expression, the barcoding approach is still unable to determine which specific LNP was responsible for the transfection event. This fundamental limitation means that the information provided by the mRNA barcode does not offer any advantage over DNA barcoding in distinguishing functional delivery contributions.

2. The mRNA barcoding approach is costly, requiring additional IVT synthesis steps and chemical modifications, making it more expensive and labor-intensive than DNA or peptide barcoding. The authors do not provide direct evidence that their method is more sensitive or superior to DNA or peptide barcoding, despite implying such an advantage. The multiplexing capability remains limited, as only three LNPs were tested in most experiments, even though they were injected at different time points. The only experiment that showcases the multiplexing potential of the mRNA barcoding strategy is Fig. 4B, which contains only 18 LNPs. This is significantly lower throughput than DNA barcoding. To substantiate the claim that the mRNA barcoding approach is superior to previous barcoding strategies, the authors should include the gold standard DNA barcoding approach as a control and directly compare detection sensitivity. A quantitative assessment of mRNA barcoding versus DNA barcoding is necessary to demonstrate that the proposed method provides more accurate and meaningful insights into mRNA delivery.

3. The authors state that their platform enables “high-resolution tracking” but fail to define the detection limit for RNA barcodes. While they reference peptide barcoding systems with a sensitivity of 0.001 mg/kg, no comparable data for RNA barcodes are presented. This omission is significant, as RNA degradation in tissues, extraction inefficiencies, and PCR/NGS biases likely affect sensitivity. Without a systematic dose-response curve spanning at least 0.0001–1.0 mg/kg, it remains unclear whether RNA barcodes achieve a lower detection limit than peptide-based systems.

4. The qPCR standard curves presented in Figure 1e were generated by spiking IVT mRNA into pre-extracted liver RNA, which bypasses RNA extraction from intact tissues. This artificially inflates sensitivity estimates by overlooking RNA loss during tissue homogenization, column retention, and enzymatic degradation. A robust standard curve should be established by intravenously injecting LNPs at varying doses, extracting RNA directly from target tissues, and correlating qPCR signals

with administered doses in vivo.

5. The study's reliance on qPCR/NGS from bulk tissue RNA (Figure 3) overlooks cell-to-cell variability and organ-specific background noise, particularly the presence of endogenous RNases in the spleen. Without quantifying false-positive rates in negative controls, the specificity of the platform remains unvalidated. The authors do not present qPCR/NGS signals in untreated mice or in mice injected with empty LNPs, making it difficult to assess background levels and specificity.

6. The claim that the platform resolves "hepatic zonal bias" and "cell-specific tropism" with high resolution is not well supported. The fluorescence microscopy images in Figures 2 and 4 show tdTomato-positive cells scattered broadly across the liver without clear zonal enrichment. No rigorous quantification of cell-specific targeting is provided. A more precise approach, such as single-cell RNA sequencing, would allow the authors to link barcodes to individual cells and offer stronger evidence of cell-specific LNP tropism.

7. The authors observed rapid LNP accumulation in target organs within 10 minutes to 1 hour and correlate this with 24-hour tdTomato expression, implying that early biodistribution determines final protein expression. However, correlation does not imply causation. Rapid accumulation indicates cellular uptake but does not confirm successful endosomal escape. To establish true functional delivery, cytosolic mRNA should be tracked via imaging or biochemical fractionation to demonstrate that early-delivered LNPs successfully release mRNA for translation.

8. Total RNA extraction with TRIzol from whole tissues cannot distinguish between endosome-trapped mRNA and functionally delivered mRNA that has escaped into the cytosol. Consequently, the proportion of mRNA that is successfully released for translation remains unknown. The authors should supplement their analysis with subcellular fractionation or fluorescence co-localization assays to confirm cytosolic mRNA release.

9. The correlation between 1-hour qPCR signals and 24-hour tdTomato fluorescence in Figure 2c, 2h, and 2m is based on limited data points, weakening the conclusion that early qPCR Ct values predict final protein expression levels. More experimental replicates with a range of mRNA doses and additional time points are needed to confirm the robustness of this correlation.

10. Figure 2d shows that liver SORT LNPs produce a high qPCR signal in the spleen at 24 hours but minimal tdTomato fluorescence, suggesting non-functional mRNA delivery. The authors attribute this discrepancy to "splenic migration," but no mechanistic data support this claim. An alternative explanation is that the qPCR signal originates from splenic macrophages, which can uptake LNPs efficiently but lack effective mRNA translation. To clarify this, the authors should sort tdTomato-positive cells and assess mRNA translation across different cell populations in the spleen.

11. The authors speculate that mRNA biodistribution at one hour correlates with functional protein expression at 24 hours. scRNA-seq should be performed to strengthen this claim at the single-cell level.

12. The authors should compare the effects of multiple injections in the same mouse with single injections in separate mice at equivalent time points. Without such a control, it remains unclear whether prior doses influence later LNP distribution and expression patterns.

13. The authors apply the method to novel LNP formulations to demonstrate its usefulness beyond known SORT LNP. It is important to demonstrate the potential of this barcoding strategy for exploring novel LNPs.

14. Key experiments, such as pharmacokinetic profiling in Figure 3k, use only two mice (n=2), which is insufficient for robust statistical analysis. Given the biological variability in LNP delivery, this small sample size raises concerns about reproducibility. Additionally, the Methods section states that RNA extraction involved centrifugation at 21,000 g for 15 minutes. The authors should verify whether this value is correct, as RNA extraction protocols typically recommend 12,000 g.

Reviewer #3

(Remarks to the Author)

This study presents a valuable advancement in tracking lipid nanoparticle (LNP) biodistribution using a multiplexed mRNA barcoding approach, enabling high-throughput, time-resolved kinetic analysis. The ability to correlate early mRNA distribution (1-hour post-injection) with functional protein expression at later time points (24-hour tdTomato fluorescence) strengthens the study's impact, providing a practical framework for future LNP screening. However, some aspects of the experimental design warrant further clarification.

1. What was the rationale for using vortex mixing instead of a microfluidic system for LNP formulation? Microfluidic systems typically provide greater reproducibility and uniformity in nanoparticle size and composition. Could the choice of vortex mixing have contributed to variability in LNP characteristics, potentially affecting biodistribution patterns?

2. The study employs Cre mRNA in a barcoding platform, yet the primary readout relies on bulk tdTomato fluorescence rather than single-cell analysis. Given that Cre-loxP systems are typically leveraged for single-cell resolution using flow cytometry, what was the rationale for using this system over a simpler luciferase-based approach, which could have provided similar insights into biodistribution? Furthermore, was there any attempt to analyze barcode distribution at the single-cell level within tissues?

3. Why were spleen data omitted in Figure 4? Given that earlier figures (e.g., Figures 2 and 3) emphasize the spleen as a key organ in LNP biodistribution, the lack of spleen analysis in Figure 4 is unexpected. Was this due to a lack of significant findings, technical limitations, or another reason? Clarifying this would strengthen the consistency of the study's conclusions.

4. In Figure 3h, the distribution pattern of Liver SORT in the liver at 1 hour, which shows similar proportions among the liver, lung, and spleen, appears quite different from Figure 2b. Don't you think that the barcoding platform, which administers multiple LNPs with different mRNAs, may not be directly comparable to the single-administration data?

Version 1:

Reviewer comments:

Reviewer #2

(Remarks to the Author)

The reviewer appreciates the authors' thoughtful revisions and acknowledges the important observation that early biodistribution kinetics (within 1 hour) strongly correlate with downstream protein expression. However, the authors did not adequately address a central concern previously raised.

Specifically, the manuscript claims that the Cre mRNA barcode design inherently links organ-level biodistribution with functional delivery. This claim remains problematic. If multiple LNPs carrying distinct Cre mRNA barcodes enter the same cell and activate tdTomato expression, the system cannot resolve which specific LNP mediated the transfection. This fundamental limitation means the current mRNA barcode approach does not offer a functional advantage over previously established DNA barcoding strategies (e.g. the one described in prior work PMID: 30275336), which enabled multiplexed Cre mRNA tracking while resolving ambiguity in the same-cell transfection.

In their rebuttal, the authors suggest that mRNA and DNA differ structurally and may result in divergent delivery profiles, citing a *Journal of Controlled Release* paper. However, they do not provide a clear mechanistic rationale or experimental evidence demonstrating how their mRNA barcode platform overcomes the critical limitation of attribution ambiguity.

To more rigorously support the platform's claims, the authors should consider conducting the following experiment: deliver mFLuc mRNA using three distinct, commercially available LNPs (e.g., MC3, ALC, and SM-102) in separate mice and quantify delivery efficiency by bioluminescence imaging. Then, pool the same LNPs, each containing a unique Cre mRNA barcode, and co-inject into a single Ai9 mouse. If the barcode sequencing results fail to differentiate the delivery efficiency of each LNP—despite all producing functional tdTomato expression—this would underscore that the current design cannot resolve relative functional contributions. This point is particularly critical given the manuscript's claim that the platform enables multiplexed LNP tracking *in vivo*.

In addition, the time-staggered multiplexed dosing protocol used in Figure 3 raises further concerns. Sequential administration of different SORT-LNPs in the same animal may induce immune responses that alter the biodistribution or uptake of subsequently delivered LNPs. Without appropriate controls, this presents a confounding variable that undermines the interpretation of kinetic data.

Supplemental experiments demonstrating that early injections do not elicit immune responses affecting later doses would help clarify the validity of the time-resolved tracking approach.

Reviewer #3

(Remarks to the Author)

I think this revised version was very well done. The response to reviewers was thoughtful, and the addition of ribosome-associated barcode profiling provided meaningful insight into the intracellular fate of mRNA. It clearly distinguished between physical delivery and functional translation, which strengthens the biological relevance of the data. I also found the use of barcoded mRNAs to optimize LNP formulations for region specific delivery, such as targeting distinct hepatic zones, particularly innovative and insightful. Overall, the revisions enhanced both the scientific depth and clarity of the manuscript.

Reviewer #4

(Remarks to the Author)

Moore et al. describe bulk sequencing to multiplex the analysis of distribution of numerous formulations of LNPs for mRNA delivery in the context of gene editing that is organ-specific (liver, lung, spleen).

Regarding Reviewer 1's technical questions, the authors partially addressed the concerns.

1. Correlation data was added for different organs, however the authors chose to merge all of the organ data together so that it was not possible to determine how protein vs. RNA correlated in the "off target" organs in SI Figure 3. These should be independently calculated so that the correlation between protein/mRNA is apparent in the different tissues.

2. Figure 2 has 3 data points shown in most graphs in panels b, g, and l, however individual data points are missing in panels c, h, and m, and it remains unclear how many replicates (mice numbers vs technical measurements) were used in each graph.

3. Figure 3 has been appropriately corrected.
4. Figure 4 appears to now show 9 data points per condition.
5. Support for the regionality of liver expression seems fine.

Regarding Reviewer 1's minor technical questions and stylistic questions, these all are addressed. However, the statistical question was insufficiently addressed as described more below.

For my personal feedback, regarding novelty and value, there is a need for advancing analytical tools for screening and evaluating LNP formulations of mRNA. The advancement in the current report is related to that of Mike Mitchell's lab (ref 18) in that mRNA barcodes were used by both reports, here with Cre recombinase instead of luciferase as the reporter gene. The authors should more clearly describe the novelty and the value of their work in this context. My major critique of this work is that it is difficult to interpret due to an underreporting of methods, insufficient statistical testing, and a lack of mechanistic considerations of classical pharmacology.

Methods. There is almost no information on animal procedures, including an institutional approval statement, methods on injections including any cannulations, euthanasia, anesthetization, restraint, groupings, ages, sexes, diets, any fasting procedures, etc. This most importantly relates to Figure 3 which appears to show 6 sequential tail vein injections in a single mouse. This is difficult using standard procedures so detailed methods should be reported. Typically, 150 μ L is the maximum intravenous volume per day recommended by AALAC and NIH. How the authors controlled volume should be described. It is also not clear if perfusion was performed and, if so, how it was performed to collect a 30 second time point. It is not clear how blood was collected and how it blood was treated, and whether it was whole blood or plasma or serum.

Statistics. There is an underuse of statistical tests for inferring relationships between data. With a small replicate number (often $N=3$), it is critical to provide p values and to use tests relevant to the data types. Pearson's R used in Figure 2 is inappropriate because (1) the data are log-incremented according to the bar graphs, (2) data points included in the correlation are approaching the maxima and minima of the y axis (0 to 100%), and (3) the data points are not homoscedastic (similar in variance). The authors may consider log-transforming their data, and if they pass homoscedasticity tests, then statistically compare the slopes across time points, rather than using R values which are nonsensical. If the data are still nonlinear, a different test for correlation should be used. Also any fitted parameters should be provided with p values to help draw conclusions. Another problem with Figure 2 is that the title and description indicate that there is a dependence of the findings on time, however there are no statistical tests comparing the two data sets at the two time points.

Other points on statistics:

1. Figure 3 and 4 are missing inferential statistics. Figure 5c does not describe the tests and comparisons. The authors should distinguish technical vs. biological replicates.
2. For Figure 2, data from micrographs are scaled by area, but data from PCR are normalized by actin, which should reflect cell number. The authors should describe how this may impact the correlation.
3. When describing a graph, the nomenclature should be "y axis title" versus "x axis title."
4. Bar graphs should not be used when plotting in log scale (doi: <https://doi.org/10.1101/2024.09.20.609464>)

Pharmacological mechanism of distribution. The most informative data shown is in Figure 3b, showing that within 30 seconds of injection, most of the material is no longer present in blood while high concentrations are found associated with tissues. This is highly unusual for colloidally stable nanoparticles in the 100 nm size range. It is a well-known phenomenon that microparticles between 1-10 microns differentially distribute to the lung (for larger particles), liver, and spleen (PMID: 7391933, 7252811). Based on these well known effects, it seems likely that the LNPs upon injection rapidly form occlusive micron-scale emboli that lodge in capillary beds. To my knowledge, there is no other known mechanism that can explain selective lung uptake within the time frame of two circulation passes in a mouse (15 s each). The authors should evaluate what happens to the administered LNPs within seconds after injection. I would expect that blood cells may be involved, possibly involving hemodynamic forces within capillaries, or this could be due to rapid LNP aggregate with blood components such as lipoproteins to form large, obstructive clusters. This is further consistent with the liver zonality, with preferential gene expression near the start of the capillary bed (periportal) which is expected to be the first region of occlusion for a microparticle.

Version 2:

Reviewer comments:

Reviewer #1

(Remarks to the Author)

The authors have provided a detailed and thoughtful response to Reviewer 4's comments and have made substantial revisions to the manuscript, figures, and supplementary data. Overall, the newly added experiments, expanded methodological descriptions, and improved statistical reporting have significantly strengthened the rigor and clarity of the study.

The only remaining comment relates to the second point concerning the distinction between this work and that of Mike Mitchell's lab (ref. 18). The current manuscript in fact advances barcoding-based screening beyond prior approaches by incorporating single-cell resolution and spatial information. These features open the door for future integration of barcoding technology with single-cell sequencing and spatial transcriptomics, which could greatly expand the impact and utility of the platform. Highlighting these possibilities more explicitly in the manuscript would further strengthen the case for novelty and

address Reviewer 4's concerns regarding conceptual advance.

Additionally, it may help reader understanding if the authors provide a reference when introducing the term "spatial flow cytometry." This will help orient readers who may be less familiar with this emerging concept. Overall, the manuscript has addressed Reviewer 4's critiques well, and emphasizing the points above would further clarify the platform's innovation and potential impact.

Reviewer #2

(Remarks to the Author)

The authors have addressed my major concerns.

1. The title of Part 4, "Barcoding streamlines new lipid discovery and formulation optimization," should be revised. I recommend deleting "new lipid discovery," as this study does not present any results related to the discovery or identification of new lipids.
2. I recommend that the authors include the gating strategy used in the liver metabolic zone study to facilitate reproducibility and make it easier for other researchers to follow the experimental workflow.

Reviewer #5

(Remarks to the Author)

RESPONSE TO REVIEWERS

Reviewer comments - Blue

Author responses – Black

Reviewer #1 (Remarks to the Author):

This manuscript by Moore et. al. presents an innovative approach utilizing multiplexed lipid nanoparticle (LNP) mRNA barcoding to enhance the characterization of tissue-specific biodistribution and functional mRNA delivery. By employing Cre recombinase mRNA barcoding and tdTomato reporter mice, the study provides a novel high-throughput method for evaluating LNP pharmacokinetics and cellular tropism across multiple organs. The authors demonstrate that barcoded LNP tracking can streamline kinetic studies, identify distinct biological outcomes, and uncover previously overlooked hepatic zonal bias and extrahepatic tropism. These findings hold substantial implications for optimizing mRNA delivery platforms for therapeutic applications. This study makes a strong contribution to the field of LNP by addressing critical challenges in pharmacokinetics evaluation. The characterization of barcoding mRNA in vivo kinetics also shows great impact on mRNA barcoding field to demonstrate mRNA have the same robustness as DNA barcoding in relatively shorter period. I recommend publication in Nature Communications after consideration of the minor questions below:

Discussion: We thank the reviewer for the kind feedback and for sharing their enthusiasm about a novel high-throughput method for evaluating LNP pharmacokinetics and cellular tropism across multiple organs.

Technical Questions:

1. Figure 2: The fluorescence images for the liver, lung, and spleen were presented, yet the correlation between tdTomato (tdTom) and qPCR was only shown for the targeted organ. To improve the robustness of the analysis, it would be beneficial to include correlation data for the liver, lung, and spleen in each organ. Ideally, for non-targeted organs, a weaker correlation would further validate the specificity of this system.

Discussion: Thank you for this comment, and yes, we agree that including additional correlation data would be valuable for readers.

Action: We included additional data in this revision. qPCR vs. tdTom correlation data in the lung, liver, and spleen for samples harvested 1 and 24 hours post-LNP administration of Liver SORT, Lung SORT, and Spleen SORT LNPs is now presented in revised **Supplementary Figure S3**. Indeed, the correlation between qPCR versus tdTom signal is weaker in non-target organs. This may be because a threshold of functional mRNA delivery is required to activate the tdTom reporter. Thank you for this excellent suggestion.

Supplementary Figure S3. Linear correlation analysis of qPCR fold changes at 1 hour or 24 hours plotted against tdTom area percentage in the liver, lung, and spleen of Ai14 mice 24 hours after administration of different doses of Liver SORT, Lung SORT, or Spleen SORT LNPs. Correlation analysis including non-target organs reveals that correlation is stronger at 1 hour than at 24 hours. N=3 image fields per point and N=3 fold change measurements per point from separate experiments.

2. Figure 2: Additional data points are needed (more than n=2) to provide stronger support for the conclusions drawn.

Action: Due to combinatorial effects of analyzing both doses and timepoints in the same experiment, the animal resource burden grows very quickly; therefore, we limited the dose analysis to two timepoints. We further expand on the conclusion that functional enrichment correlates with biodistribution more strongly at earlier timepoints in **Figure 3** and **Supplementary Figure S3-5**. To further strengthen this point, we purified ribosomal fractions from a barcoded kinetics experiment and validated that barcodes from earlier time points were enriched in ribosome-associated fractions. These results have been added to (**Supplementary Figure S5g-i**).

Supplementary Figure 5 | Validation and imaging of barcode readouts and early LNP biodistribution. (g-i) Multiplexed barcode kinetics experiment showing time-coded barcodes associated with total RNA or ribosome profiled RNA. Early biodistribution and enrichment can result in functional delivery

3. Figure 3: There appears to be an inconsistency between the curve plots and the bar graphs. For instance, in Figure 3f, the liver SORT data shows a peak level at approximately 10 minutes, whereas in Figure 3h, the peak level for the spleen is at 3 hours. A similar inconsistency is observed between Figures 3f and 3j. The authors should clarify these discrepancies.

Discussion: Because the stacked graphs were plotted as the normalized distribution across the tested organs (i.e., the percentage of total signal at the indicated timepoints), the y-axis values depend on both the sum of total signal from other organs and the signal within an organ. For this reason, peak signal within an organ does not always coincide with peak signal as a percent of total. We apologize that the initially chosen graph type did not clearly convey the findings. We understand this concern, and have adjusted the graph type accordingly.

Action: Based on feedback from you and all reviewers, we have re-plotted these graphs as separated bar charts and changed the y-axis to reflect the absolute fold change normalized to the LNP input pool, $\frac{2^{-(CT\ barcode_N - CT\ actin)}}{2^{-CT\ barcode_N} / \sum_i^n 2^{-CT\ barcodes}}$ where the numerator is from harvested samples and the denominator is from the LNP pooled input (to account for slight variations in barcode input concentration and qPCR efficiency) (see revised **Figure 3**). Please note that the data and conclusions remain unchanged. We appreciate this comment and believe that the revised plots more clearly convey the conclusions of the data sets.

Figure 3. (h-j) Barcoded qPCR was used to map the distribution of Liver SORT, Lung SORT, and Spleen SORT LNPs in the lung, liver, and spleen at each injection timepoint. **(h)** Liver SORT LNPs primarily distribute to the liver between 5 minutes and 1 hour but also accumulate in the spleen. Most Liver SORT barcodes were localized to the spleen by 3 hours. **(i)** Lung SORT LNPs rapidly and preferentially distribute to the lung, with modest and relatively slower liver and spleen accumulation. **(j)** Spleen SORT LNPs primarily distribute to the liver and spleen between 5 minutes and 1 hour.

4. Figure 4b and 4c: Additional data points are needed in both figures to provide stronger support for the conclusions drawn.

Discussion: Sure, additional data points can be added to support the conclusions drawn.

Action: We added tdTom quantification replicates and additional panels with a correlation analysis between tdTom vs. barcodes to revised **Figure 4**.

Figure 4. (b) Normalized NGS barcode counts in the Liver, Lung, Spleen, and Kidney following multiplexed administration of the noted barcoded LNPs. LNPs were formulated with multiple distinct Cre mRNA barcodes, pooled, and administered as a pool IV to N=3 mice at 0.01 mg/kg/formulation; 0.08 mg/kg total dose and collected 1 hour after administration. **(c)** Linear correlation analysis of the normalized barcode counts plotted against the normalized tdTom area fraction for each individually administered formulation in the lung, liver, spleen, and kidney. N=3 image fields quantified per formulation.

5. The conclusion that liver SORT LNPs are restricted to periportal hepatocytes while LNPs containing 10% DOTAP expand distribution to pericentral hepatocytes could be better supported. Consider addressing: 1) The second row of images in Figure 4a appears vague. A high-resolution image is necessary for better visualization. 2) Quantification of distribution in pericentral hepatocytes is required. Flow cytometry analysis may be needed to better quantify tdTomato+ GS+ hepatocytes, which would strengthen the conclusion.

Discussion: Thank you for these suggestions. We believe that quantitative insights into hepatic zonal distribution of LNPs is one of the key broader impacts of the manuscript. We completed a significant number of additional experiments for this revision to both improve the quantification and expand the data sets.

The liver is organized into repeating, roughly hexagonal units termed lobules that consist of a central vein flanked by six portal veins. Both the gene expression and metabolic function of hepatocytes along the portal-to-central axis are spatially distinct, leading to a phenomenon called liver metabolic zonation. The extreme pericentral region expresses the marker gene Glutamine Synthetase (GS), while E-cadherin (Ecad) expression forms a gradient along the portal (highest) to central (lowest) axis¹ (**Figure 5a**).

Action: To add support to this finding, we quantified overlap between GS and tdTom signal in higher resolution fluorescence images of sectioned livers. A new chart has been added as **Figure 5c** to detail the results. Additionally, we performed a new analysis showing representative line profiles and accompanying images of GS and tdTom signal extending from a GS+ central vein to a neighboring portal

vein in Liver SORT LNP and DOTAP10 LNP treated mouse livers **Figure 5d**. This analysis shows the zonal bias of Liver SORT LNPs vs. DOTAP10. These mice were given a slightly lower dose (0.1 mg/kg Cre mRNA) to enable more precise quantification of the differences. Finally, we performed a new barcoded multiplexed experiment using the same Liver SORT and DOTAP LNP series as previously included in the initial submission. Following IV administration, we fractionated the liver by surface Ecad expression, which is graded from portal tract high to central vein low, using FACS¹. Liver SORT LNP barcodes were enriched in the Ecad-high periportal population and DOTAP 10 LNP barcodes were enriched in the Ecad-low pericentral population (**Figure 5e**). This analysis showed that these LNPs can exhibit zonal bias.

Figure 5. Spatial mapping of LNP activity across liver zonation. (a) Schematic illustrating the liver architecture at increasing magnification: whole liver, liver lobule with a central vein surrounded by six portal triads, and a central-to-portal axis with glutamine synthetase (GS)-positive cells near the central vein and Ecad-positive cells near the portal tract. (b) Reporter imaging reveals LNP activity across metabolic zones in response to DOTAP titration. Liver SORT transfects primarily periportal regions, sparing GS+ pericentral cells. 10% DOTAP inclusion enhances transfection in GS+ cells, while higher DOTAP levels (20–50%) reduce liver targeting. (c) Quantification of GS and reporter signal colocalization highlights zonal transfection biases of each formulation. (d) Fluorescent line scans over

central-to-portal axis show no tdTom GS overlap when Liver SORT LNPs are dosed at 0.1 mg/kg and stochastic distribution with GS overlap when 10% DOTAP LNPs are given at the same dose. (e) Multiplexed barcode analysis with flow cytometric sorting by Ecad expression shows barcode enrichment patterns: Liver SORT enriches in Ecad high expressing cells (periportal); both formulations are represented in Ecad mid-expressing (mid-lobular) population, and 10% DOTAP LNPs are enriched in the Ecad low (pericentral) expressing cells.

Minor Technical Questions:

1. In Figure 2, although the biodistribution of Lung SORT LNPs was lung-centered, there was still a lower correlation of qPCR results to 24h data. Why?

Discussion: Across both the 1 hour and 24 hour time points, correlation analyses reveal acceptable linearity. However, we acknowledge that there is some deviation from the linear trend in the tdTom response and measured fold changes at the extreme ends of the dose range. Biologically, this could be due to 1) saturation of accumulation/elimination mechanisms 2) a thresholding/saturation effect for tdTom expression. While non-linear models also fit the data well with more degrees of freedom, the overall conclusion is the same, fold change values measured at 1 hour are a better fit for the tdTom response.

Action: We added a sentence to the Main Text to introduce this possibility. “We noted that the tdTom response had some deviation from linearity at the extreme ends of the dose range, possibly due to an activation threshold apparent at the lowest dose, and saturation at 1 mg/kg. However, the conclusion that 1 hour qPCR measurements share stronger correlation to the observed protein response remained true regardless of the curve fitting parameters.”

2. Please also include spleen in Figure 4 / supplementary figure as it is a major organ relevant to validating the robustness of the Cre mRNA barcoding system.

Action: Certainly. We added spleen to our analyses of the LNP series in **Figure 4**.

3. While all the microscopy images are striking, consider including some quantification of the signal in these images to further support the conclusions made about the kinetics of Cre mRNA signal vs Cre mRNA accumulation.

Action: Great idea! Image quantifications have been added to **Figure 2** and **Figure 4**.

4. Please include citations for why you think immune cells are trafficking from the liver to the spleen. Has this been observed before?

Discussion: While beyond the goals of this work, our speculation that immune cells may be one source of splenic accumulation is based on the observation that the splenic mRNA signal tends to be more stable

than in tissues like the liver, lung, or blood. In some cases, we observed a modest increase in splenic signal that occurred after the initial accumulation phase (**Figure 4f**), despite rapid blood clearance. Previous work with exosomes and macrophage depletion studies have shown that nanoparticles are rapidly sequestered in tissues by immune populations², and it stands to reason that a component of spleen accumulation, and non-functional distribution in particular, could be mediated by an immune response. We think it would be interesting to study this interaction further.

Action: An additional reference² reviewing this point have been added to the revised Main Text.

5. As we now know, protein corona around LNPs is an important determining characteristic in their tropism. With this in mind, why do you think you observe kidney tropism with addition of DOTAP in your LNP?

Discussion: This is a very insightful comment. Yes, in fact, in recent publications,³⁻⁶ we have characterized the protein corona of SORT LNPs and correlated receptor expression to SORT LNP uptake. Our data showed that DOTAP SORT LNPs adsorb Vitronectin from plasma to subsequently internalize in $\alpha V\beta 3$ integrin (Vitronectin receptor, Vtn-R) expressing cells. For example, in a recent manuscript focused on genome editing in the lungs, we carefully quantified Vtn-R expression across all mouse tissues (Figure 2 and Figure S18 in Ref.⁶) and carefully studied on- and off-target editing in target and non-target organs including the kidneys. You are right that there is a strong foundation upon which the newly discovered (via barcoded SORT LNPs for the first time) kinetics and cellular tropism data rests.

Action: We added a sentence to the main to introduce the endogenous targeting mechanism of DOTAP LNPs “Mechanistically, this phenomenon is at least partially explained by an endogenous targeting mechanism, whereby DOTAP LNPs adsorb Vitronectin from plasma to subsequently internalize in $\alpha V\beta 3$ integrin (Vitronectin receptor, Vtn-R) expressing cells that are enriched in the lung”, and a sentence referencing discovery of VTN-R expression in the kidneys “Recent profiling of $\alpha V\beta 3$ integrin (Vitronectin receptor, Vtn-R) expression throughout mouse tissues found that populations of cells within the kidney express Vtn-R.”

We also expanded the references and discussion in the Main Text to highlight the connection between protein corona, receptor expression, and cellular internalization.

Statistical Questions:

1. Statistical tests are largely missing from the current studies. Please add statistical analyses to all studies possible to show any significance.

Discussion: Sure, additional statistical analyses are warranted.

Action: We performed additional replicates to verify data sets, and added statistics to relevant quantifications. In **Figure 2**, **Figure S2**, and **Figure S3**, Pearson correlation coefficients were computed for each dose response. In **Figure 4b**, ANOVA was performed to compare the normalized barcode counts

between formulations in each organ. In **Figure 4c**, Pearson correlation analysis was used. In **Figure 5c**, AVOVA was used and are plotted.

Stylistic Recommendations:

1. Although Figure 1e shows near-perfect correlation, consider moving it to the supplementary information as information about mRNA modifications does not seem to add too much value to your discussion about multiplexed barcoding and validation.

Discussion: Sure, we are open to moving the figure panels if supported by the editor.

Action: We moved **Figure 1e** to **Supplementary Figure S2a**.

2. Consider rephrasing “Cre mRNA signal was detectable within the first 3 hours” to “Cre mRNA levels in tissue were detectable within 3 hours” as signal implies fluorescence or some other kind of visual signal.

Discussion: Thank you for the suggestion! We agree that the suggested rephrasing more clearly conveys the detection of the mRNA transcript.

Action: The text now reads “Although detectable out to 72 hours, the majority of transcript abundance was observed within the initial 3 hours.”

3. In the section “Multiplexed barcoded PK...”, please clarify what “study mice” are.

Action: We intended for this phrase to reference mice from the experiment; however, to avoid potential confusion, we agree and have removed the phrase “study mice” from the text.

4. In figure 3, please clarify the x axes as 00:05 is not in hours. Also indicate where the 30 second timepoint is.

Discussion: We agree that the notation was confusing.

Action: We revised the axis label to (hh:mm:ss) and added timepoint labels.

Reviewer #2 (Remarks to the Author):

The study by Moore et al. presents an mRNA barcoding strategy for LNP screening at the tissue level. The authors designed an mRNA barcode encoding Cre recombinase, which triggers tdTomato expression in Ai14 mice, allowing for visualization and enrichment of functional delivery events. The barcoded mRNA enables multiplexed LNP tracking, with its distribution quantified using NGS and qPCR. The study examines the biodistribution of Selective Organ Targeting (SORT) LNPs in the liver, lung, and spleen over time. However, many critical issues limit the novelty and scientific impact of this work. First, the mRNA barcoding design closely follows a previously published mRNA barcode in the Journal of Controlled Release (PMID: 31678653), with the only modification being the replacement of the ORF from luciferase to Cre recombinase. This represents only a minor conceptual advance over prior work, as the study does not introduce any new LNP formulations or provide mechanistic insights into LNP design and function. The authors argue that DNA barcodes, being separate from mRNA, could lead to decoupled biodistribution and altered LNP properties, but this hypothesis was not experimentally tested. The claim remains speculative, especially given that prior studies have successfully used DNA barcoding to screen hundreds of LNPs in a single animal, whereas the current mRNA barcoding approach was only demonstrated with three types of LNPs in most of its studies. Notably, the barcoding approach did not uncover any previously unknown mechanisms of LNP delivery. The only new finding (Liver SORT LNPs primarily went to periportal hepatocytes in Fig. 4A) was identified using standard fluorescence imaging in mice individually dosed with single LNP formulations, rather than through multiplexed barcoding. This contradicts the claimed utility of the barcoding approach. The study also does not identify any new LNP formulations with improved properties and instead applies the barcoding technique to pre-existing SORT LNPs. The study primarily confirms already expected biodistribution patterns rather than providing new discoveries about LNP behavior or design principles. While the barcoding approach may be useful for small-scale LNP kinetic studies, its practical advantages over prior barcoding methods remain unproven. Overall, this study provides an incremental methodological refinement rather than a significant conceptual or technical progress. It does not convincingly demonstrate an advantage over existing DNA or peptide barcoding methods. The lack of essential controls and sensitivity validation as well as the failure to confirm cytosolic mRNA release limit the impact of this work.

Discussion: Thank you for the feedback. We have considered the points carefully, and respond below detailing improvements and additions to the manuscript to address the concerns.

1. The authors claim that the Cre mRNA barcode design inherently links organ-level biodistribution with functional delivery; however, this is not entirely accurate. If multiple LNPs containing distinct Cre mRNA barcodes enter the same cell and successfully induce tdTomato expression, the barcoding approach is still unable to determine which specific LNP was responsible for the transfection event. This fundamental limitation means that the information provided by the mRNA barcode does not offer any advantage over DNA barcoding in distinguishing functional delivery contributions.

Discussion: Yes, your premise is true, biodistribution and functional delivery cannot be distinguished directly at the barcode level – the connection can only correlate biodistribution and function with further measurement of tdTom expression. However, the conclusion that mRNA barcodes offer no advantage over DNA barcodes is not supported by previous studies.⁷ The most pressing application of LNPs currently includes mRNA vaccines for infectious diseases and cancer, mRNA based protein replacement, and mRNA-encoded genome editors.⁸⁻¹⁰ Because short oligomeric DNA does not share the same physical properties of mRNA that is typically >100 longer and single stranded, studies have shown that different nanoparticle designs are required.¹¹ The challenges of DNA versus mRNA delivery have been reviewed, including from the perspective of physics and chemistry.^{12, 13} Therefore, we reasoned that mRNA barcodes are more useful to evaluate delivery events of mRNA LNPs than DNA barcodes because the encapsulated cargo recapitulates the same cargo type with its unique physiochemical properties.⁷

In addition, DNA barcodes can only reveal biodistribution, as there is no coupled functional readout. So while DNA barcodes may be useful to study biodistribution of DNA LNPs (where the target nucleic acid drug is DNA), DNA barcodes are less applicable for studying mRNA LNPs (where the target nucleic acid drug is mRNA).

In the mRNA barcode paper published in the *Journal of Controlled Release* (PMID: 31678653)⁷ cited in the reviewer comments above, the authors did study and compare mRNA and DNA barcodes. The authors found that delivery profiles reported by mRNA vs. DNA barcodes were not well-correlated:

Fig. 6. Encapsulation of barcoded DNA (b-DNA) versus b-mRNA in LNPs alters *in vivo* delivery. (A) 16 LNP formulations used in this study were now used to each encapsulate unique b-DNA instead of b-mRNA. b-DNA contained universal primer sites, a 10-nucleotide barcode sequence, and a 10-nucleotide UMI region to minimize polymerase chain reaction (PCR) bias. (B–C) 16 b-DNA LNP formulations were pooled (1 μ g b-DNA per injection for each formulation) and administered to C57BL/6 mice intravenously. 4 h post injection, b-DNA delivery to the liver (B) and spleen (C) was quantified. N = 4 mice per group. (D–E) *In vivo* delivery of 16 b-mRNA LNP formulations was plotted against the delivery of 16 b-DNA LNP formulations. Method to calculate b-DNA delivery is explained in detail in the experimental section. R² values were calculated based on a linear regression model. Data were plotted as mean \pm SD.

Furthermore, the same study also found that DNA barcodes correlated to functional delivery with reduced accuracy as compared to mRNA barcodes:

Fig. 7. Comparison of b-mRNA system versus b-DNA system to predict functional mRNA delivery *in vivo* (A, B) b-mRNA LNP delivery was plotted against luciferase expression in the liver (A) and spleen (B) of luciferase mRNA LNP-treated mice. (C, D) Similarly, b-DNA LNP delivery was plotted against luciferase expression in the liver (C) and spleen (D) of luciferase mRNA LNP-treated mice. Data were plotted as mean \pm SD.

Consequently, we feel that mRNA barcoding offers distinct advantages that allowed determination of LNP pharmacokinetics and cellular tropism across multiple organs in the current manuscript.

Action: The Main Text has been updated.

2. The mRNA barcoding approach is costly, requiring additional IVT synthesis steps and chemical modifications, making it more expensive and labor-intensive than DNA or peptide barcoding. The authors do not provide direct evidence that their method is more sensitive or superior to DNA or peptide barcoding, despite implying such an advantage.

Discussion: As noted above and reported by others,⁷ DNA barcodes do not correlate well with functional mRNA delivery outcomes, and thus are poorly suited for discovery and pharmacokinetic evaluation of mRNA LNPs. We agree that DNA is less expensive than mRNA, however, the trade off in accuracy has limitations for making rigorous conclusions.

Peptide barcodes are less sensitive and more expensive than mRNA barcodes.^{14, 15} Because peptide barcodes are mRNA encoded, a similar cost and labor is required to generate peptide barcode libraries^{14, 15} (both are mRNA based). Downstream analysis of the peptide (protein) products produced by mRNA encoded (peptide) barcodes is achieved using semi-quantitative mass spectrometry. This mass spectrometry method of detection is expensive, labor intensive, and less sensitive than RNA sequencing. Furthermore, peptide stability and detection is a significant optimizable parameter that may require

additional efforts. In terms of sensitivity, RNA libraries have been generated from single cells, whereas proteomic libraries require much more starting material. Further, peptide barcodes have not been successfully implemented outside of the liver in published literature to date.

While implementation of DNA barcodes is significantly less expensive and labor intense, the quality of the data may require additional secondary screens to examine functional activity. DNA barcodes would report DNA LNP internalization in cells/tissues, but cannot report key functional steps including endosomal escape and functional delivery (e.g. mRNA translation, pDNA transcription, genome editing, etc.). Further, some barcode-based studies are less relevant/not possible with DNA barcodes, such as the pharmacokinetic profiling we performed here that report on mRNA fate or barcoded ribosome profiling.

For these reasons, we believe that mRNA barcodes are functionally useful, most appropriate to study mRNA LNPs designed to deliver therapeutic mRNAs, and worth the cost and effort associated with their use.

The multiplexing capability remains limited, as only three LNPs were tested in most experiments, even though they were injected at different time points. The only experiment that showcases the multiplexing potential of the mRNA barcoding strategy is Fig. 4B, which contains only 18 LNPs. This is significantly lower throughput than DNA barcoding. To substantiate the claim that the mRNA barcoding approach is superior to previous barcoding strategies, the authors should include the gold standard DNA barcoding approach as a control and directly compare detection sensitivity. A quantitative assessment of mRNA barcoding versus DNA barcoding is necessary to demonstrate that the proposed method provides more accurate and meaningful insights into mRNA delivery.

Discussion: Yes, we understand these points, and we agree that DNA barcodes hold key advantages including cost, sensitivity, and ease of use. We fully support this strategy and agree that it can be helpful in nanoparticle and other applications (e.g., DNA-labeled small molecule libraries).

Regarding the request to compare DNA barcodes versus mRNA barcodes, this work has been undertaken and published previously in the *Journal of Controlled Release* (PMID: 31678653).⁷ Consequently, it is not necessary to repeat these expensive experiments herein, given the significant funding constraints that we and other NIH-funded labs are currently going through with recent grant cancellations and cuts.

Action: We further refined the Main Text to capture advantages of the DNA barcode approach.

3. The authors state that their platform enables “high-resolution tracking” but fail to define the detection limit for RNA barcodes. While they reference peptide barcoding systems with a sensitivity of 0.001 mg/kg, no comparable data for RNA barcodes are presented. This omission is significant, as RNA degradation in tissues, extraction inefficiencies, and PCR/NGS biases likely affect sensitivity. Without a systematic dose-response curve spanning at least 0.0001–1.0 mg/kg, it remains unclear whether RNA barcodes achieve a lower detection limit than peptide-based systems.

Discussion: This is an insightful comment that we appreciate very much.

Action: In this revised manuscript, we went ahead and performed additional experiments to test the lower limits of mRNA barcode detection. We expanded the dose response from 0.0001-1.0 mg/kg, as proposed. Using Liver SORT LNPs, we completed the broader dose response and harvested tissue at 24 hours. We added the results to new **Supplementary Figure S2**. The lowest doses were readily detectible by PCR at 24 hours.

Supplementary Figure 2 | Standardization and dose-response analysis of liver-targeting LNP formulations. (a) A standard curve generated by spiking known masses of barcoded mRNA into total RNA, followed by reverse transcription and qPCR, reveals a linear relationship between spike-in RNA mass and qPCR fold change. (b) A dose-response experiment using 0.0001–1.0 mg/kg of barcoded mRNA-loaded LNPs shows a strong linear correlation between administered dose and liver qPCR signal at 24 hours post-injection. (c) Reporter fluorescence images from livers collected in the dose-response study.

4. The qPCR standard curves presented in Figure 1e were generated by spiking IVT mRNA into pre-extracted liver RNA, which bypasses RNA extraction from intact tissues. This artificially inflates sensitivity estimates by overlooking RNA loss during tissue homogenization, column retention, and enzymatic degradation. A robust standard curve should be established by intravenously injecting LNPs at varying doses, extracting RNA directly from target tissues, and correlating qPCR signals with administered doses *in vivo*.

Discussion: During the initial testing and validation of the barcoding platform, we performed the spike-in experiment to build confidence that PCR based detection was a reliable and linear quantification method for IVT mRNA in the absence of delivery and extraction variables that may be non-linear.

All of the dose response curves in **Figure 2**, and the dose response curve added to **Supplementary Figure S2b**, were obtained by extracting LNP delivered mRNAs directly from the tissues.

Action: We apologize that this was unclear in the original submission. We have revised the experimental section to more carefully describe the procedures.

5. The study's reliance on qPCR/NGS from bulk tissue RNA (Figure 3) overlooks cell-to-cell variability and organ-specific background noise, particularly the presence of endogenous RNases in the spleen. Without quantifying false-positive rates in negative controls, the specificity of the platform remains unvalidated. The authors do not present qPCR/NGS signals in untreated mice or in mice injected with empty LNPs, making it difficult to assess background levels and specificity.

Discussion: Thank you for the careful reading and comments on background levels and specificity.

We do include PBS injected mice in the workflow. The NGS experimental workflow requires a PCR amplification step that yields no product in the PBS condition. Regarding specificity, in a recent barcode sequencing experiment that yielded ~50M total reads, >99% of the reads were mapped to a barcode used in the experiment, while no reads mapped to unused barcodes. While the potential for PCR amplification and sequencing errors to produce reads that map to a real barcode but vary by a single base exist, these make up <1% of total reads and are stochastic across the barcode library. Additionally, by requiring barcode counts to have an exact barcode match, we can ensure that no false positives are counted, however negligible.

For qPCR experiments, we used a PBS control to calculate the relative fold change $2^{-(\Delta\Delta CT)}$ values instead of the absolute fold change $2^{-(\Delta CT)}$ values presented here. We found that in all cases, the Cre probe CT values in PBS controls were above 30 regardless of organ (usually 33-38) while treated animals ranged from 15-29 depending on timing and dose. Therefore, absolute fold changes in PBS conditions were essentially zero: $2^{-(35-actin)} \approx 0$. The result was very large relative fold changes, but nearly identical curve shapes, R^2 values, and overall conclusions. We preferred to retain the biological meaning of absolute fold changes, which give approximately the detection of Cre mRNA (or Cre barcodes) as compared to Actin mRNA. Below is a sample of relative vs. absolute qPCR data.

Figure R1. qPCR results from the liver, lung, and spleen after administration of Cre mRNA loaded Liver SORT LNPs. The table shows CT values for delivered Cre mRNA and endogenous Actin mRNA. Absolute and relative fold changes were calculated and plotted side by side to compare dose correlation and distribution. The analyses have identical R^2 values, and nominal differences in overall distribution.

6. The claim that the platform resolves “hepatic zonal bias” and “cell-specific tropism” with high resolution is not well supported. The fluorescence microscopy images in Figures 2 and 4 show tdTomato-positive cells scattered broadly across the liver without clear zonal enrichment. No rigorous quantification of cell-specific targeting is provided. A more precise approach, such as single-cell RNA sequencing, would allow the authors to link barcodes to individual cells and offer stronger evidence of cell-specific LNP tropism.

Discussion: Thank you for these suggestions. We believe that quantitative insights into hepatic zonal distribution of LNPs is one of the key broader impacts of the manuscript. We completed a significant number of additional experiments for this revision to both improve the quantification and expand the data sets.

The liver is organized into repeating, roughly hexagonal units termed lobules that consist of a central vein flanked by six portal veins. Both the gene expression and metabolic function of hepatocytes along the portal-to-central axis are spatially distinct, leading to a phenomenon called liver zonation. The extreme pericentral region expresses the marker gene Glutamine Synthetase (GS), while E-cadherin (Ecad) expression forms a gradient along the portal (highest) to central (lowest) axis (**Fig 5a**).

Action: The manuscript has been expanded from 4 figures to 5 figures, where the additional experiments are captured in new **Figure 5**. To add support to this finding, we quantified overlap between GS and tdTom signal in higher resolution fluorescence images of sectioned livers. A new chart has been added

as **Figure 5c** to detail the results. Additionally, we performed a new analysis showing representative line profiles and accompanying images of GS and tdTom signal extending from a GS+ central vein to a neighboring portal vein in Liver SORT LNP and DOTAP10 LNP treated mouse livers **Figure 5d**. This analysis shows the zonal bias of Liver SORT LNPs vs. DOTAP10. These mice were given a lower dose (0.1 mg/kg Cre mRNA) to enable more precise quantification of the differences. Finally, we performed a new barcoded multiplexed experiment using the same Liver SORT and DOTAP LNP series as previously included in the initial submission. Following IV administration, we fractionated the liver by surface Ecad expression, which is graded from portal tract high to central vein low, using FACS. Liver SORT LNP barcodes were enriched in the Ecad-high periportal population and DOTAP 10 LNP barcodes were enriched in the Ecad-low pericentral population (**Figure 5e**). This analysis showed that these LNPs can exhibit zonal bias.

Figure 5 | Spatial mapping of LNP activity across liver zonation. (a) Schematic illustrating the liver architecture at increasing magnification: whole liver, liver lobule with a central vein surrounded by six portal triads, and a central-to-portal axis with glutamine synthetase (GS)-positive cells near the central

vein and Ecad-positive cells near the portal tract. **(b)** Reporter imaging reveals LNP activity across metabolic zones in response to DOTAP titration. Liver SORT transfects primarily periportal regions, sparing GS+ pericentral cells. 10% DOTAP inclusion enhances transfection in GS+ cells, while higher DOTAP levels (20–50%) reduce liver targeting. **(c)** Quantification of GS and reporter signal colocalization highlights zonal transfection biases of each formulation. **(d)** Fluorescent line scans over central-to-portal axis show no tdTom GS overlap when Liver SORT LNPs are dosed at 0.1 mg/kg and stochastic distribution with GS overlap when 10% DOTAP LNPs are given at the same dose. **(e)** Multiplexed barcode analysis with flow cytometric sorting by Ecad expression shows barcode enrichment patterns: Liver SORT enriches in Ecad high expressing cells (periportal); both formulations are represented in Ecad mid-expressing (mid-lobular) population, and 10% DOTAP LNPs are enriched in the Ecad low (pericentral) expressing cells.

7. The authors observed rapid LNP accumulation in target organs within 10 minutes to 1 hour and correlate this with 24-hour tdTomato expression, implying that early biodistribution determines final protein expression. However, correlation does not imply causation. Rapid accumulation indicates cellular uptake but does not confirm successful endosomal escape. To establish true functional delivery, cytosolic mRNA should be tracked via imaging or biochemical fractionation to demonstrate that early-delivered LNPs successfully release mRNA for translation.

Discussion: We agree, and we further appreciate this comment in the context above discussing DNA, mRNA, and protein barcoding approaches with their unique advantages and disadvantages.

This point goes to the crux of why we completed the experiments reported herein. We started with the naïve question “when should we harvest our samples?” In early experiments, we harvested samples at 24 hours in accordance with most prior literature reports.^{7, 14-19} This time point did not reveal expected enrichment versus known protein expression (**Supplementary Figure S3a**). This prompted us to perform a kinetics experiment (**Supplementary Figure S4d**), which surprisingly revealed early accumulation and rapid subsequent elimination. Follow-up biodistribution studies performed at 1 hour (**Supplementary Figure S3b**) were, interestingly, a much better match to the functional outcomes. These and subsequent observations have led us to conclude that, in many cases, early biodistribution shares more correlation with functional outcomes than later timepoints. This finding has not been reported before to our knowledge, and is a key take home message since past barcoding studies have only examined time points of 4 hours or later post-injection, detection of LNP delivery in tissues may have been missed.¹⁶ While we do not argue that these delivery events are all functional, our work does reveal that mRNA LNPs accumulate in tissues more rapidly than previously known, and that functional efficacy correlates more strongly with early barcode time points.

Action: The points made here are important, and prompted us to use unique aspects of mRNA barcodes to reveal deeper insights at the cellular level. In response to this suggestion, we performed ribosome profiling of barcoded multiplexed kinetics samples using a well established ribosome pulldown assay.²⁰ As in **Figure 3**, barcoded LNPs were given at different timepoints to the same mouse. Bulk tissues were dissociated to single cells in the presence of the protein translation inhibitor cycloheximide to stall

ribosomes on actively translating transcripts, then ribosomes were immunoprecipitated on magnetic beads, along with bound transcripts. Quantifying the ribosome bound barcodes that were administered at different time points enabled us to reconstruct the kinetics of ribosomal engagement for the delivered mRNA. This experiment revealed a time delay between peak biodistribution and peak ribosomal engagement, however it also showed that early mRNA biodistribution is associated with functional mRNA delivery. Note that the curves are normalized, not absolute scale, such that all the timepoints within each curve sum to 1. Results have been added to **(Supplementary Fig S5g-i)**

Supplementary Figure 5 | Validation and imaging of barcode readouts and early LNP biodistribution. (g-i) Multiplexed barcode kinetics experiment showing time-coded barcodes associated with total RNA or ribosome profiled RNA. Early biodistribution and enrichment can result in functional delivery.

8. Total RNA extraction with TRIzol from whole tissues cannot distinguish between endosome-trapped mRNA and functionally delivered mRNA that has escaped into the cytosol. Consequently, the proportion of mRNA that is successfully released for translation remains unknown. The authors should supplement their analysis with subcellular fractionation or fluorescence co-localization assays to confirm cytosolic mRNA release.

Discussion: We agree. The use of bulk analysis for validation was a practical consideration as fractionation is technically challenging and time-consuming. However, our above mentioned ribosome profiling experiment was meant to tackle the question of functional kinetics.

Action: In addition to the ribosome profiling experiment, we have also added confocal microscopy of liver sections following Liver SORT delivery of Cy5 labeled mRNA (**Supplementary Figure S5j**). We co-labeled the images using an antibody for eIF4e, which is an intracellular protein that binds the 5' mRNA cap as part of the mRNA translation initiation process. The confocal setup allowed us to optically section the tissue into very thin sections so that only intracellular signal is detected. Some of the intracellular Cy5 mRNA signal appears to colocalize with the co-labeled eIF4e. While we do not attempt to quantify endosomal escape with these images, they show that some fraction of LNP delivered mRNA is cytosolic, even at this early timepoint.

j Liver SORT Liver 20 min post injection confocal ~1.5 μ M optical section

Supplementary Figure 5 | Validation and imaging of barcode readouts and early LNP biodistribution. (j) Confocal imaging of liver tissue 20 minutes post-LNP administration shows fluorescently labeled mRNA localized within cells, co-stained with the capping protein eIF4E.

9. The correlation between 1-hour qPCR signals and 24-hour tdTomato fluorescence in Figure 2c, 2h, and 2m is based on limited data points, weakening the conclusion that early qPCR Ct values predict final protein expression levels. More experimental replicates with a range of mRNA doses and additional time points are needed to confirm the robustness of this correlation.

Action: We added additional replicates and statistics to these analyses. Because the 24 hour tdTom functional readouts are shared between qPCR timepoints, the kinetics analyses in **Figure 3** serve as additional timepoints. We do not claim that early biodistribution is predictive, only correlative. Critically, biodistribution is necessary but insufficient for function. We contend that the biodistribution snapshot at early timepoints is more correlative to function for the LNPs tested here.

Figure 2 | LNP-mRNA biodistribution and functional tropism demonstrates time-dependent correlation. (a) Endogenous Rosa-tdTom fluorescent micrographs of Ai14 mouse liver, lung, and spleen 24 hours post-intravenous (IV) administration of Liver SORT LNPs carrying 0.1, 0.25, or 1.0 mg/kg Cre recombinase shows liver-specific protein production. (b) tdTom dose response image quantification. (c) RT-qPCR performed on liver RNA extracts from mice that received Liver SORT LNPs at various doses (d) Linear regression analysis comparing 1 hour qPCR signal vs. 24 hour tdTom signal shows strong correlation. (e) Linear regression analysis of 24 hour qPCR signal vs. 24 hour tdTom signal. Correlation

is linear, but weaker than at 1 hr. **(f)** Endogenous Rosa-tdTom fluorescent micrographs of Ai14 mouse liver, lung, and spleen 24 hours post-intravenous administration of Lung SORT LNPs carrying 0.1, 0.25, or 1.0 mg/kg Cre mRNA **(g)** tdTom dose response image quantification. **(h)** RT-qPCR performed on lung RNA extracts from mice that received Lung SORT LNPs at various doses. **(j)** Linear regression analysis of 1 hour qPCR signal versus tdTom signal shows strong correlation. **(j)** Linear regression analysis of 24 hour qPCR signal vs. 24 hour tdTom signal shows reduced correlation. **(k)** Endogenous Rosa-tdTom fluorescent micrographs of Ai14 mouse liver, lung, and spleen 24 hours post-intravenous administration of Spleen SORT LNPs carrying 0.1, 0.25, or 1.0 mg/kg of Cre recombinase mRNA shows spleen-specific protein production. **(l)** tdTom dose response image quantification. **(m)** RT-qPCR performed on spleen RNA extracts from mice that received Spleen SORT LNPs at various doses. **(n)** Linear regression analysis of 1 hour qPCR signal vs. 24 hour tdTom signal shows strong correlation. **(o)** Linear regression analysis of 24 hour qPCR signal vs. 24 hour tdTom signal. Correlation is linear, but weaker than at 1 hr. (b,g,l: N=3 image fields) (c, h, m; n=3 mice per dose) (a, f, k; Scale bars: 200 μm .)

10. Figure 2d shows that liver SORT LNPs produce a high qPCR signal in the spleen at 24 hours but minimal tdTomato fluorescence, suggesting non-functional mRNA delivery. The authors attribute this discrepancy to “splenic migration,” but no mechanistic data support this claim. An alternative explanation is that the qPCR signal originates from splenic macrophages, which can uptake LNPs efficiently but lack effective mRNA translation. To clarify this, the authors should sort tdTomato-positive cells and assess mRNA translation across different cell populations in the spleen.

Discussion: During platform validation, we observed that the splenic mRNA qPCR signal tends to be more stable than in tissues like the liver, lung, or blood. In some cases, we observed a modest increase in splenic signal that occurred hours after the initial peak accumulation phase, despite rapid blood clearance. Further, the 24 hour accumulation of Liver SORT and Lung SORT LNPs in the spleen showed much less dose dependence than in other organs.

There are several potential mechanistic explanations, and we agree that the source of the qPCR signal could be due to macrophage uptake without functional delivery. Thank you for your thoughtful mechanistic questions. We believe that they warrant further investigation that lies beyond the scope of the work contained herein. Previously, our lab has produced follow-on mechanistic studies to fully and robustly characterize the biological interactions of LNPs with organ-targeting outcomes,^{3, 5, 6, 21} and we plan on carrying that forward with the systems studied in this work. Conducting these studies with sufficient depth will take significant time and effort. In addition to the spleen, Lung SORT LNPs accumulate in the liver but are weakly functional there, further suggesting that Lung SORT LNPs cannot escape the endosome of liver cells. In the future, we anticipate conducting an array of experiments, including mRNA barcodes, to consider transcytosis, tight junction passage, macrophage-mediated transfer, exosome-mediated transfer, protein corona analyses, and studies in genetically engineered mice as examples. We nevertheless remain interested in these mechanistic questions and will continue to work on this topic for future publications.

Action: Here “migration” was used to refer to the changing proportion of spleen localized signal over time rather than an actual movement of individual LNPs. We removed the term ‘migration’ from the figure legends and corrected our messaging to reflect that LNP accumulation and degradation kinetics in the spleen appear more complicated than the liver or lung.

11. The authors speculate that mRNA biodistribution at one hour correlates with functional protein expression at 24 hours. scRNA-seq should be performed to strengthen this claim at the single-cell level.

Discussion: We show that, for SORT LNPs, early biodistribution and enrichment share more correlation to downstream protein production than later biodistribution and enrichment. By enrichment, we mean the tendency for an LNP to preferentially distribute quickly to organs with associated functional activity versus LNPs without functional activity. While we agree that scRNA sequencing could yield interesting data using the barcoding platform, we are unfortunately unable to perform these experiments at the current time due to acute funding limitations stemming from cuts at our university. We feel that the significant added cost of scRNA sequencing is not justified in the context of this work, as the current conclusions can be made with next generation sequencing (NGS).

Action: We added the results of a 1 hour vs. 24 hour NGS experiment (**Supplementary Figure S3m**)

Supplementary Figure 3 | Comparative biodistribution and reporter correlation of tissue-targeting SORT LNPs. (m) Heatmap summarizing a pooled barcoding experiment in which barcoded Liver, Lung, and Spleen SORT LNPs were co-administered and harvested at 1 h or 24 h. Target organ enrichment was markedly higher at 1 h for each formulation, while enrichment declined substantially by 24 h.

We also added additional emphasis to the discussion around this point, and its limitations to our tested LNPs. The following point was emphasized in the revised discussion section “It is worth noting that the specific mRNA accumulation, elimination, and degradation kinetics reported here could be influenced by a variety of factors such as the base modifications, m7g cap1 structure, untranslated regions, or polyA tail, among others, used in this study. Additionally, further screening must be performed to generalize LNP kinetics to a broader range of chemistries beyond SORT formulations. Studying generalizability to other mRNA modifications and nanoparticle carriers will be an important future goal.”

12. The authors should compare the effects of multiple injections in the same mouse with single injections in separate mice at equivalent time points. Without such a control, it remains unclear whether prior doses influence later LNP distribution and expression patterns.

Discussion: To minimize potential effects from multiple injections, each LNP was dosed at 0.01 mg/kg, and each pooled injection was given in 50 μ L total volume. Thus, 6 injections added 300 μ L of circulatory volume and totaled 0.18 mg/kg. These values are consistent with a low dose single injection.

Action: To validate that the pharmacokinetics of multiplexed injections are equivalent to single injections when given in low doses and small volumes, we compared the results of a 72 hour time course in single or multiplex. The results have been added to new **Figure S5** and show that multiplexed injections have similar kinetics to single injections. Further, we compared the distributions between the 0.01 mg/kg single injection group from the 1-hr dose response data to the 1 hour multiplexed kinetics distribution. We found that the distributions were not significantly different between single and multiplexed injections.

Supplementary Figure 5 | Validation and imaging of barcode readouts and early LNP biodistribution. (b–c) Multiplexed and single-injection kinetic studies at matched timepoints yield comparable barcode profiles and similar biodistribution patterns for liver-, lung-, and spleen-targeting LNPs.

13. The authors apply the method to novel LNP formulations to demonstrate its usefulness beyond known SORT LNP. It is important to demonstrate the potential of this barcoding strategy for exploring novel LNPs.

Discussion: In this manuscript, we report the development of an LNP barcoding approach that enables multiplexed cell-type specific LNP tracking *in vivo* using unique barcoded mRNA. We demonstrate that coupling between LNP biodistribution and downstream protein production occurs at unexpectedly early timepoints due to rapid organ accumulation. This surprising observation is currently unknown publicly,

and herein discovered and reported for the first time. This insight allowed us to identify LNPs with extrahepatic activity and zoned liver delivery profiles. Although the platform can be used in the future to potentially discover new LNPs with novel properties, the enclosed data set is nevertheless a significant advance in LNP discovery and characterization for several reasons:

1. Systematic kinetic studies: Our barcoding platform enables the high-resolution analysis of mRNA accumulation and degradation kinetics in various organs, streamlining the identification of optimal LNP formulations.
2. Cell-specific LNP characterization: By using our barcoding strategy, we can precisely characterize LNP delivery efficiency to specific cell types within target organs, enhancing the understanding of cell-specific delivery dynamics.
3. Liver zoned delivery: Our study characterized nanoparticles with hepatic zonal bias, providing insights into the spatial distribution of LNPs within the liver and optimizing targeted delivery. Zoned delivery has not been observed or achieved before to our knowledge, yet is very intriguing mechanistically and for therapeutic applications.
4. Distinguishing similar formulations: The barcoding system allows us to differentiate subtle yet significant variations within a series of similar LNP formulations, ensuring the identification of the most effective candidates.

14. Key experiments, such as pharmacokinetic profiling in Figure 3k, use only two mice (n=2), which is insufficient for robust statistical analysis. Given the biological variability in LNP delivery, this small sample size raises concerns about reproducibility. Additionally, the Methods section states that RNA extraction involved centrifugation at 21,000 g for 15 minutes. The authors should verify whether this value is correct, as RNA extraction protocols typically recommend 12,000 g.

Action: Thank you for this observation. Additional replicates have been added throughout the manuscript. Thank you for catching the centrifugation typo.

Reviewer #3 (Remarks to the Author):

This study presents a valuable advancement in tracking lipid nanoparticle (LNP) biodistribution using a multiplexed mRNA barcoding approach, enabling high-throughput, time-resolved kinetic analysis. The ability to correlate early mRNA distribution (1-hour post-injection) with functional protein expression at later time points (24-hour tdTomato fluorescence) strengthens the study's impact, providing a practical framework for future LNP screening. However, some aspects of the experimental design warrant further clarification.

Discussion: Thank you very much for the helpful feedback! We are excited to resubmit a revised manuscript that provides further clarification of the findings.

1. What was the rationale for using vortex mixing instead of a microfluidic system for LNP formulation? Microfluidic systems typically provide greater reproducibility and uniformity in nanoparticle size and composition. Could the choice of vortex mixing have contributed to variability in LNP characteristics, potentially affecting biodistribution patterns?

Discussion: We appreciate this point and completely agree. Recognizing the need to standardize LNP preparation and characterization methodologies, we published a paper in 2023 in *Nature Protocols* to provide readers with a step-by-step guide to formulate, characterize, and assay *in vitro* and *in vivo* SORT LNPs.²² We provide protocols to prepare 4A3-SC8- and DLin-MC3-DMA-based Liver, Lung, and Spleen SORT LNPs via three technical methods: Pipette mixing, vortex mixing, and microfluidic mixing. This was done to provide readers with options to cover small-, medium-, and large-scale production. Notably, our optimized methodologies demonstrated that the distribution of LNP activity did not change with different mixing methods.²² While we agree that microfluidic mixing offers certain advantages, it is a bottleneck for manufacturing many different LNPs at once. Thus, for pooled barcode and related studies, we utilized the vortex mixing method that enabled higher throughput. We employed dynamic light scattering (DLS) (for size and uniformity) and Ribogreen (RNA encapsulation) assays to ensure that LNP physical properties fell within the standards of the field. We agree that the precise values reported here could vary slightly between mixing methods, however we believe that the overall conclusions would apply equally.

Action: We added the following sentence to the main text “We chose to utilize vortex mixing because this method is compatible with high throughput, automatable workflows, and produces high quality LNPs with comparable *in vivo* properties to microfluidic mixing.²²”

2. The study employs Cre mRNA in a barcoding platform, yet the primary readout relies on bulk tdTomato fluorescence rather than single-cell analysis. Given that Cre-loxP systems are typically leveraged for single-cell resolution using flow cytometry, what was the rationale for using this system over a simpler luciferase-based approach, which could have provided similar insights into

biodistribution? Furthermore, was there any attempt to analyze barcode distribution at the single-cell level within tissues?

Discussion: Yes, we appreciate this point. Our goal for this current paper was to leverage the mRNA barcode system for cell-specific insights that are not possible using readouts such as luciferase. The use of bulk analysis in some areas was a practical consideration during platform validation as single cell analysis is costly and time-consuming. Furthermore, the bulk behavior of the tested LNPs was largely known and therefore provided a reference dataset to validate the performance of the barcodes. Insights including LNP zonal bias were found using the tdTom readout in combination with fluorescence microscopy that would not have been possible with luciferase alone.

Action: In response to this suggestion, we performed FACS-based barcoding using the tdTom reporter to fractionate tdTom positive and negative cells. Additionally, in the liver, we separated cells by the zoned surface marker E-cadherin (Ecad). This enabled us to confirm that LNPs can exhibit liver zonal bias at the biodistribution level using barcode data. Furthermore, the set of LNPs enriched in tdTom positive sorted cells differed significantly from the tdTom negative population. These experiments showcase the enhanced resolution potential of this platform.

Figure 4 | (d) Flow cytometry dot plots with tdTomato expression on the x-axis from the lung, spleen, and kidney of mice that received the barcode pool. (e) Heat map showing the normalized barcode distribution in tdTomato positive or negative cells isolated using FACS. Barcodes in the tdTomato positive population are enriched for more functionally active formulations vs. tdTomato negative cells.

Figure 5 | Spatial mapping of LNP activity across liver zonation. (e) Multiplexed barcode analysis with flow cytometric sorting by Ecad expression shows barcode enrichment patterns: Liver SORT enriches in Ecad high expressing cells (periportal); both formulations are represented in Ecad mid-expressing (mid-lobular) population, and 10% DOTAP LNPs are enriched in the Ecad low (pericentral) expressing cells.

3. Why were spleen data omitted in Figure 4? Given that earlier figures (e.g., Figures 2 and 3) emphasize the spleen as a key organ in LNP biodistribution, the lack of spleen analysis in Figure 4 is unexpected. Was this due to a lack of significant findings, technical limitations, or another reason? Clarifying this would strengthen the consistency of the study's conclusions.

Discussion: We initially omitted the spleen because it was not a target organ for the set of tested LNPs. We also thought that including too much data might confuse readers. However, we see your point here and totally agree.

Action: We added a panel of fluorescent spleen images from the series and included the barcoding results of bulk spleen, tdTom positive spleen, and tdTom negative spleen in the revised manuscript.

Figure 4 | (a) Endogenous tdTom fluorescent micrographs of the liver, lung, spleen and kidneys of mice administered 0.5 mg/kg Cre mRNA encapsulated in the noted formulations. (b) Multiplexed pooled barcode experiment shows the normalized distribution of barcodes in the liver, lung spleen, and Kidney. LNP pool contained 3 unique barcodes per DOTAP formulation and 2 unique barcodes per Liver SORT formulation; N=3 mice received the pool. For statistics see Supplementary Table 2.

4. In Figure 3h, the distribution pattern of Liver SORT in the liver at 1 hour, which shows similar proportions among the liver, lung, and spleen, appears quite different from Figure 2b. Don't you think that the barcoding platform, which administers multiple LNPs with different mRNAs, may not be directly comparable to the single-administration data?

Discussion: For the referenced comparison, it is only relevant to compare the 0.01 mg/kg single injection data point as this is the dose for the kinetics experiment. While less lung signals were initially detected in the single injection experiment, the proportions in the liver and spleen were comparable. Replicate experiments have shown these different methods to be within experimental error for the referenced comparison.

Action: We added two new panels to **Supplementary Figure S5** that compare single injection to multiplex injections. Both the distribution and kinetics findings were similar between these approaches. Thank you for this suggestion!

Supplementary Figure 5 | Validation and imaging of barcode readouts and early LNP biodistribution. (b–c) Multiplexed and single-injection kinetic studies at matched timepoints yield comparable barcode profiles and similar biodistribution patterns for liver-, lung-, and spleen-targeting LNPs.

References

1. Ben-Moshe, S. et al. Spatial sorting enables comprehensive characterization of liver zonation. *Nat. Metab.* **1**, 899–911 (2019).
2. Cataldi, M., Vigliotti, C., Mosca, T., Cammarota, M. & Capone, D. Emerging role of the spleen in the pharmacokinetics of monoclonal antibodies, nanoparticles and exosomes. *Int. J. Mol. Sci.* **18** (2017).
3. Dilliard, S.A., Cheng, Q. & Siegwart, D.J. On the mechanism of tissue-specific mRNA delivery by selective organ targeting nanoparticles. *Proc. Natl. Acad. Sci. U.S.A.* **118**, e2109256118 (2021).
4. Vaidya, A. et al. Expanding RNAi to kidneys, lungs, and spleen via selective organ targeting (SORT) siRNA lipid nanoparticles. *Adv. Mater.* **36** (2024).
5. Dilliard, S.A. et al. The interplay of quaternary ammonium lipid structure and protein corona on lung-specific mRNA delivery by selective organ targeting (SORT) nanoparticles. *J. Control. Release* **361**, 361–372 (2023).
6. Sun, Y. et al. In vivo editing of lung stem cells for durable gene correction in mice. *Science* **384**, 1196–1202 (2024).

7. Guimaraes, P.P.G. et al. Ionizable lipid nanoparticles encapsulating barcoded mRNA for accelerated in vivo delivery screening. *J. Control. Release* **316**, 404–417 (2019).
8. Kazemian, P. et al. Lipid-nanoparticle-based delivery of CRISPR/Cas9 genome-editing components. *Mol. Pharm.* **19**, 1669–1686 (2022).
9. Jacob, E.M., Huang, J. & Chen, M. Lipid nanoparticle-based mRNA vaccines: a new frontier in precision oncology. *Precis. Clin. Med.* **7**, pbae017 (2024).
10. Lu, R.M. et al. Current landscape of mRNA technologies and delivery systems for new modality therapeutics. *J. Biomed. Sci.* **31**, 89 (2024).
11. Scholz, C. & Wagner, E. Therapeutic plasmid DNA versus siRNA delivery: common and different tasks for synthetic carriers. *J. Control. Release* **161**, 554–565 (2012).
12. Hou, X., Zaks, T., Langer, R. & Dong, Y. Lipid nanoparticles for mRNA delivery. *Nat. Rev. Mater.* **6**, 1078–1094 (2021).
13. Miller, J.B. & Siegwart, D.J. Design of synthetic materials for intracellular delivery of RNAs: From siRNA-mediated gene silencing to CRISPR/Cas gene editing. *Nano Research* **11**, 5310–5337 (2018).
14. Odunze, U. et al. RNA encoded peptide barcodes enable efficient in vivo screening of RNA delivery systems. *Nucleic Acids Res.* **52**, 9384–9396 (2024).
15. Rhym, L.H., Manan, R.S., Koller, A., Stephanie, G. & Anderson, D.G. Peptide-encoding mRNA barcodes for the high-throughput in vivo screening of libraries of lipid nanoparticles for mRNA delivery. *Nat. Biomed. Eng.* **7**, 901–910 (2023).
16. Dahlman, J.E. et al. Barcoded nanoparticles for high throughput in vivo discovery of targeted therapeutics. *Proc. Natl. Acad. Sci. U.S.A.* **114**, 2060–2065 (2017).
17. Sago, C.D. et al. High-throughput in vivo screen of functional mRNA delivery identifies nanoparticles for endothelial cell gene editing. *Proc. Natl. Acad. Sci. U.S.A.* **115**, E9944–E9952 (2018).
18. Dobrowolski, C. et al. Nanoparticle single-cell multiomic readouts reveal that cell heterogeneity influences lipid nanoparticle-mediated messenger RNA delivery. *Nat. Nanotechnol.* **17**, 871–879 (2022).
19. Sago, C.D. et al. Nanoparticles that deliver RNA to bone marrow identified by in vivo directed evolution. *J. Am. Chem. Soc.* **140**, 17095–17105 (2018).
20. Metz, J.B. et al. High-throughput translational profiling with riboPLATE-seq. *Sci. Rep.* **12**, 5718 (2022).
21. Lian, X.Z. et al. Bone-marrow-homing lipid nanoparticles for genome editing in diseased and malignant haematopoietic stem cells. *Nat. Nanotechnol.* **19**, 1409–1417 (2024).
22. Wang, X. et al. Preparation of selective organ targeting (SORT) lipid nanoparticles (LNPs) using multiple technical methods for tissue-specific mRNA delivery. *Nat. Protoc.* **18**, 265–291 (2022).

RESPONSE TO REVIEWERS

Reviewer comments - Blue

Author responses – Black

Reviewer #2 (Remarks to the Author):

The reviewer appreciates the authors' thoughtful revisions and acknowledges the important observation that early biodistribution kinetics (within 1 hour) strongly correlate with downstream protein expression. However, the authors did not adequately address a central concern previously raised.

Specifically, the manuscript claims that the Cre mRNA barcode design inherently links organ-level biodistribution with functional delivery. This claim remains problematic. If multiple LNPs carrying distinct Cre mRNA barcodes enter the same cell and activate tdTomato expression, the system cannot resolve which specific LNP mediated the transfection. This fundamental limitation means the current mRNA barcode approach does not offer a functional advantage over previously established DNA barcoding strategies (e.g. the one described in prior work PMID: 30275336), which enabled multiplexed Cre mRNA tracking while resolving ambiguity in the same-cell transfection.

In their rebuttal, the authors suggest that mRNA and DNA differ structurally and may result in divergent delivery profiles, citing a Journal of Controlled Release paper. However, they do not provide a clear mechanistic rationale or experimental evidence demonstrating how their mRNA barcode platform overcomes the critical limitation of attribution ambiguity.

To more rigorously support the platform's claims, the authors should consider conducting the following experiment: deliver mFLuc mRNA using three distinct, commercially available LNPs (e.g., MC3, ALC, and SM-102) in separate mice and quantify delivery efficiency by bioluminescence imaging. Then, pool the same LNPs, each containing a unique Cre mRNA barcode, and co-inject into a single Ai9 mouse. If the barcode sequencing results fail to differentiate the delivery efficiency of each LNP—despite all producing functional tdTomato expression—this would underscore that the current design cannot resolve relative functional contributions. This point is particularly critical given the manuscript's claim that the platform enables multiplexed LNP tracking *in vivo*.

Discussion: Thank you for the feedback, and continued recommendations to strengthen the manuscript. We appreciate your recognition of the correlation between early biodistribution kinetics and downstream protein expression. This observation remains a key finding that has not been previously published. We acknowledge your concern regarding attribution ambiguity in our Cre mRNA barcode system and agree that our platform is unable to resolve single-cell attribution in cases of co-transfection. Instead, it enables comparisons of mRNA delivery across LNPs in a multiplexed format. We agree that this limitation should be clearly stated and have revised the manuscript accordingly.

Regarding the cited DNA barcoding strategy (PMID: 30275336), we note that while it offers valuable insights, it similarly relies on correlation between barcode abundance and functional readouts, without resolving attribution when multiple constructs co-transfect the same cell.

We agree with your recommendation that performing an additional experiment (as detailed above) to add data would more rigorously support the platform's claims. We have completed the experiment as noted below, and the results show that the Cre mRNA barcode design can differentiate the delivery efficiency of each LNP.

Action: We revised the manuscript to explicitly clarify the scope and limitations of our platform. Additionally, we conducted the experiment proposed by the reviewer to rigorously test our system's ability to differentiate functional delivery across LNPs:

- We delivered mFLuc mRNA via three distinct LNPs (ALC, SM-102, LP01) in separate mice and quantified bioluminescence to establish baseline delivery efficiency.
- We co-injected the same LNPs, each carrying a unique Cre mRNA barcode, into a single Ai14 mouse.
- We compared barcode abundance in tdTomato⁺ tissues with the prior mFLuc delivery profiles.

The results showed that barcode abundance in tdTomato⁺ tissues reflected the luciferase-based delivery profiles observed in separate mice. Specifically, the LNPs that produced stronger luciferase bioluminescence signals also yielded proportionally higher Cre barcode counts in the co-injected Ai14 mice. These results confirm that our platform can differentiate the relative delivery efficiency of each LNP *in vivo* under multiplexed conditions.

Supplementary Figure 6. Cre mRNA barcodes recapitulate the functional activity of commercial LNPs. (a) *Ex vivo* images of mouse livers 6 hours after LNP delivery of firefly luciferase mRNA. LNPs

were loaded with 0.1 mg/kg of firefly luciferase mRNA formulated using commercially available ionizable lipids (IL) and the following molar ratios of helper lipids: (IL:DSPC:Cholesterol:DMG-PEG2000); ALC-0315 (46.3:9.4:42.7:1.6), SM-102 (50:10:38.5:1.5), or LP01 (45:9:44:2). (b) Total radiance quantification of *ex vivo* livers showed that ALC-0315 LNPs resulted in the most luciferase protein production, followed by LP01 and SM-102 based LNPs. (c) In parallel, established LNPs were loaded with Cre mRNA barcodes, pooled together, and administered to Ai14 tdTom reporter mice at 0.01 mg/kg RNA per barcode (n=3 mice). The barcode readouts from livers collected 1-hour after multiplexed LNP administration in all three pooled mice agreed with the luminescence results obtained using individual mice.

In addition, the time-staggered multiplexed dosing protocol used in Figure 3 raises further concerns. Sequential administration of different SORT-LNPs in the same animal may induce immune responses that alter the biodistribution or uptake of subsequently delivered LNPs. Without appropriate controls, this presents a confounding variable that undermines the interpretation of kinetic data.

Discussion: We recognize that biological responses to LNPs could potentially confound interpretation of sequential kinetics experiments. This is an important consideration for validating time-resolved tracking. To mitigate this potential, we administered low doses of pooled LNPs, 0.01 mg/kg of mRNA per formulation/timepoint, totaling 0.18 mg/kg over 3 hours. We believe these low doses do not saturate biological uptake mechanisms and are well tolerated.

Action: We compared kinetic traces in the liver, measured using qPCR, from sequential barcoded injections to those established from individual mice. We found that the curves were not significantly different. We further compared the biodistributions measured at 1 hour in mice that received only 1 injection to those from mice that received sequential barcoded injections and found no significant difference. These data have been added to **Supplemental Figure 4b,c** to clarify the validity of our time-resolved tracking approach. These results indicate that sequential administration of different SORT LNPs in the same animal does not alter the biodistribution or uptake of subsequently delivered LNPs.

Supplementary Figure 5 | Validation and imaging of barcode readouts and early LNP biodistribution. (b–c) Multiplexed and single-injection kinetic studies at matched timepoints yield comparable barcode profiles and similar biodistribution patterns for liver-, lung-, and spleen-targeting LNPs.

Additionally, we collected histology in major organs from animals that received 1.0 mg/kg of mRNA loaded into SORT LNPs. We did not observe obvious signs of an inflammatory response 24 hours after LNP administration. These data are now included in **Supplemental Figure 8**, and further support that sequential administration of different SORT LNPs in the same animal does not alter the biodistribution or uptake of subsequently delivered LNPs.

Supplementary Figure 8 | SORT LNPs are well tolerated. H&E images from major organs 24 hours after receiving 1.0 mg/kg RNA loaded in SORT LNPs. No obvious signs of injury or inflammation were apparent.

Thank you again for your constructive feedback! We believe these additional experiments and clarifications enhance the manuscript's rigor and transparency.

Reviewer #3 (Remarks to the Author):

I think this revised version was very well done. The response to reviewers was thoughtful, and the addition of ribosome-associated barcode profiling provided meaningful insight into the intracellular fate of mRNA. It clearly distinguished between physical delivery and functional translation, which strengthens the biological relevance of the data. I also found the use of barcoded mRNAs to optimize LNP formulations for region specific delivery, such as targeting distinct hepatic zones, particularly innovative and insightful. Overall, the revisions enhanced both the scientific depth and clarity of the manuscript.

Discussion: We sincerely thank you for your generous and encouraging feedback. We are especially grateful for the recognition of the ribosome-associated barcode profiling, which we believe offers a critical layer of insight into the intracellular fate of mRNA beyond biodistribution. Distinguishing physical delivery from functional translation was a central goal of our revision, and we are pleased that this addition strengthened the biological relevance of our findings.

We also appreciate your comments on the use of barcoded mRNAs to optimize LNP formulations for region-specific delivery, including hepatic zonation. This application reflects our broader aim to develop tools that not only characterize delivery but also guide rational design for spatial targeting *in vivo*.

Thank you again for your thoughtful review and support of the revised manuscript!

Reviewer #4 (Remarks to the Author):

Moore et al. describe bulk sequencing to multiplex the analysis of distribution of numerous formulations of LNPs for mRNA delivery in the context of gene editing that is organ-specific (liver, lung, spleen).

Regarding Reviewer 1's technical questions, the authors partially addressed the concerns. 1. Correlation data was added for different organs, however the authors chose to merge all of the organ data together so that it was not possible to determine how protein vs. RNA correlated in the "off target" organs in SI Figure 3. These should be independently calculated so that the correlation between protein/mRNA is apparent in the different tissues.

Discussion: We agree that merging organ data obscures tissue-specific relationships between mRNA and protein expression. We initially plotted the data this way because no reporter signal, and therefore no relationship to qPCR signal, was observed in some off target organs. Correlating the aggregate was an attempt to quantify the relationship we perceive when analyzing at the data; while we do see biodistribution to organs without corresponding functional activity, the magnitude of on-target qPCR signal in early measurements is much greater.

Action: In response, we have recalculated and plotted the correlation data for each organ independently in the revised **Supplementary Figure 3**. Additionally, all correlation results and curve fitting parameters

were added to the associated data file. In addition, we added normalized biodistribution and functional distribution data to this figure to more clearly visualize the key message.

Supplementary Figure 3 | Comparative biodistribution and reporter correlation of tissue-targeting SORT LNPs. (a-c) tdTom reporter quantification from mice that received various doses of Cre mRNA packaged in Liver SORT (a), Lung SORT (b), or Spleen SORT (c) LNPs. **(d-i)** tdTom

reporter signal plotted against qPCR fold changes in off-target organs at 1 h or 24 h following administration of Liver SORT (**d, g**) Lung SORT (**e, h**), and Spleen SORT (**i, l**). *Did not pass homoscedasticity test. (**j-l**) Biodistribution profiles at multiple doses for Liver SORT (**j**), Lung SORT (**k**), and Spleen SORT (**l**) LNPs; early timepoints align more closely with observed reporter expression. (**m-n**) Heatmap summarizing a pooled barcoding experiment in which barcoded Liver, Lung, and Spleen SORT LNPs were co-administered and harvested at 1-hour (**m**) or 24-hours (**n**). Target organ enrichment was markedly higher at 1 h for each formulation, while enrichment declined substantially by 24 h.

2. Figure 2 has 3 data points shown in most graphs in panels b, g, and I, however individual data points are missing in panels c, h, and m, and it remains unclear how many replicates (mice numbers vs technical measurements) were used in each graph.

Discussion: We appreciate your attention to data transparency.

Action: We have revised the figure legends to explicitly state the number of biological replicates (mice) and technical replicates (e.g., PCR wells, imaging fields) for each panel. Graphs in **Figure 2** were updated and all graphs now display individual data points where applicable.

3. Figure 3 has been appropriately corrected.

4. Figure 4 appears to now show 9 data points per condition.

5. Support for the regionality of liver expression seems fine.

Discussion: Thank you very much for checking these revisions addressing the prior comments from other reviewers!

Regarding Reviewer 1's minor technical questions and stylistic questions, these all are addressed. However, the statistical question was insufficiently addressed as described more below.

For my personal feedback, regarding novelty and value, there is a need for advancing analytical tools for screening and evaluating LNP formulations of mRNA. The advancement in the current report is related to that of Mike Mitchell's lab (ref 18) in that mRNA barcodes were used by both reports, here with Cre recombinase instead of luciferase as the reporter gene. The authors should more clearly describe the novelty and the value of their work in this context.

Discussion: We appreciate your comparison to prior work from Mitchell et al. (ref 18). While both platforms use mRNA barcoding, our approach differs in several key respects:

- We use Cre recombinase as a binary functional reporter, enabling cell and tissue-level attribution of successful transfection.
- Our ribosome-associated barcode profiling distinguishes physical delivery from translation, adding mechanistic depth.

- We demonstrate regional targeting within organs (e.g., hepatic zonation), which extends the utility of barcoding beyond organ-level resolution.

Action: We have revised the Introduction and Discussion to more clearly articulate these distinctions and the unique contributions of our platform.

My major critique of this work is that it is difficult to interpret due to an underreporting of methods, insufficient statistical testing, and a lack of mechanistic considerations of classical pharmacology.

Methods. There is almost no information on animal procedures, including an institutional approval statement, methods on injections including any cannulations, euthanasia, anesthetization, restraint, groupings, ages, sexes, diets, any fasting procedures, etc. This most importantly relates to Figure 3 which appears to show 6 sequential tail vein injections in a single mouse. This is difficult using standard procedures so detailed methods should be reported. Typically, 150 μ L is the maximum intravenous volume per day recommended by AALAC and NIH. How the authors controlled volume should be described. It is also not clear if perfusion was performed and, if so, how it was performed to collect a 30 second time point. It is not clear how blood was collected and how it blood was treated, and whether it was whole blood or plasma or serum.

Action: We agree that detailed methodological reporting is essential. The revised Methods section now includes:

- Institutional approval statements.
- Injection procedures, including tail vein volumes, restraint methods, and anesthesia protocols.
- Mouse demographics (strain, age, sex, diet, fasting status).
- Blood collection details and clarification that whole blood was used for analysis.

The *Animal models* section of the methods section now reads:

“Animal experiments were approved by the Institution Animal Care and Use Committee of The University of Texas Southwestern Medical Center, and were consistent with local, state, and federal regulations as applicable. B6.Cg-Gt(ROSA)26Sortm14(CAG-tdTomato)Hze/J (also known as Ai14) mice were obtained from The Jackson Laboratory (007914) and bred to maintain homozygous expression of the Cre reporter allele that has a loxP-flanked STOP cassette preventing transcription of a CAG promoter-driven red fluorescent tdTomato protein. Following Cre-mediated recombination Ai14 mice express tdTomato fluorescence. All animals were maintained on a 12/12-h light/dark schedule at a mean temperature of 22°C and fed with ad libitum normal chow. All experiments were performed in at least 3 biological replicates. Male and female mice age 6-12 weeks were used for experiments. Age and sex were distributed equally across experimental groups.

Lateral tail vein injections were performed under physical restraint. Sequential tail vein injections for multiplexed kinetics study were performed as follows: (1) Each injection contained 0.01 mg/kg/formulation, 0.03 mg/kg total mRNA dissolved in 50 μ L PBS. (2) Injections at 3h, 1h, and 30m

timepoints were administered under physical restraint sequentially in one lateral tail vein (distal to proximal), while 10m, 5m, and 30s timepoints were delivered under isoflurane anesthesia in the opposite vein, also distal to proximal. (3) Following the last injection, mice were immediately euthanized by cervical dislocation. (4) Blood collection was immediately performed via cardiac puncture using a 25G needle inserted into the left ventricle until ~100 uL was drawn. Whole blood was used for downstream analysis by directly adding 1 mL of TRIzol and following the RNA extraction procedure described above. (5) Major organs were immediately collected in the following order: heart, lung, liver, spleen, and kidneys. Utensils were briefly wiped between organs to avoid cross contamination. No perfusion was used prior to collecting tissues; tissue samples were added directly to TRIzol and processed as described for RNA extraction. (6) Replicate injection times were staggered to facilitate handling only one mouse at a time.”

Regarding the sequential injections in **Figure 3**, while we recognize and regularly observe the guidance that total injection volume per mouse does not exceed 150 μ L per day, here we administered the barcoded LNPs over six 50 μ L injections totaling 300 μ L. This volume is below the 25 μ L/g (~600 uL) maximum daily injection volume put forth by PMID: 11180276 and recommended by IQ Consortium for slow IV bolus injections given over 5-10 minutes (here 3 hours).

Statistics. There is an underuse of statistical tests for inferring relationships between data. With a small replicate number (often N=3), it is critical to provide p values and to use tests relevant to the data types. Pearson’s R used in Figure 2 is inappropriate because (1) the data are log-incremented according to the bar graphs, (2) data points included in the correlation are approaching the maxima and minima of the y axis (0 to 100%), and (3) the data points are not homoscedastic (similar in variance). The authors may consider log-transforming their data, and if they pass homoscedasticity tests, then statistically compare the slopes across time points, rather than using R values which are nonsensical.

If the data are still nonlinear, a different test for correlation should be used. Also any fitted parameters should be provided with p values to help draw conclusions.

2. Figure 2 has 3 data points shown in most graphs in panels b, g, and I, however individual data points are missing in panels c, h, and m, and it remains unclear how many replicates (mice numbers vs technical measurements) were used in each graph.

Discussion: We appreciate your attention to data transparency and thoughtful critique regarding the use of Pearson’s correlation in **Figure 2**. To clarify, all datasets for which correlation was assessed were first evaluated for homoscedasticity and all passed without requiring further transformation. The data were originally plotted on a log scale solely to facilitate visualization across a wide dynamic range, not to imply log-transformed statistical treatment.

We acknowledge your concern regarding potential boundary effects near 0% and 100%. However, the distribution of data points remained sufficiently dispersed across the range, and variance was consistent across the measured domain. Given these conditions, Pearson’s R remains an appropriate metric for assessing linear association in this context.

Action: To avoid confusion, we have updated the figure to display data on linear axes. We have added clarification to the figure legend and methods section to explicitly state that the data were not transformed and that homoscedasticity was confirmed prior to correlation analysis. We have also included all fitting parameters in the associated data file for each curve. The language surrounding the correlation and time-dependencies was softened and clarified to more accurately reflect our observations. This chapter of the results section now follows the following format:

“Liver SORT LNPs drove efficient, dose-dependent tdTom expression in the liver, with minimal signal in the spleen across the dose range. Some tdTom expression was observed in splenic red pulp at 1.0 mg/kg, possibly indicating hepatic saturation at this dose. No tdTom signal was detected in the lungs at any dose (**Fig. 2a, Supplementary Fig. S3a**). Cre mRNA levels measured by qPCR showed strong dose dependence at both 1h and 24h, but were ~100-fold higher at 1h across the dose range (**Fig. 2b,c**). Liver tdTom expression correlated with liver Cre mRNA levels at both timepoints, however the correlation was modestly stronger at 1h (**Fig. 2d,e**). Moreover, the distribution of Cre mRNA across the liver, lung, and spleen shared more resemblance to the tdTom distribution when measured at 1h (primarily liver) than at 24h (primarily spleen) (**Supplementary Fig. S3l**). These results demonstrated that the biodistribution of Liver SORT LNPs measured 1h post-injection were a better match for the observed functional activity in the liver, while the biodistribution at 24h showed potentially misleading splenic tropism.”

We have also revised the figure legends to explicitly state the number of biological replicates (mice) and technical replicates (e.g., PCR wells, imaging fields) for each panel.

Figure 2 | LNP-mRNA biodistribution correlates with functional activity (a) Endogenous Rosa-tdTom fluorescent micrographs of Ai14 mouse liver, lung, and spleen 24 hours post-intravenous (IV) administration of Liver SORT LNPs carrying 0.1, 0.25, or 1.0 mg/kg Cre recombinase shows liver-specific protein production. (b,c) Cre qPCR performed on liver, lung, and spleen 1-hour (b) or 24-hours (c) after administration of Liver SORT LNPs at various doses. (d,e) Linear regression analysis comparing tdTom signal vs. 1-hour qPCR (d) or 24-hour qPCR (e) shows linear correlation that is modestly stronger at 1-hour. Pearson's R calculated after **testing homoscedasticity in Graphpad Prism**. (f) Endogenous Rosa-tdTom fluorescent micrographs of Ai14 mouse liver, lung, and spleen 24 hours post-intravenous (IV) administration of Lung SORT LNPs carrying 0.1, 0.25, or 1.0 mg/kg Cre recombinase. (g,h) Cre

qPCR performed on liver, lung, and spleen 1-hour (**g**) or 24-hours (**h**) after administration of Lung SORT LNPs at various doses. (**i,j**) Linear regression analysis comparing tdTom signal vs. 1-hour qPCR (**i**) or 24-hour qPCR (**j**) shows linear correlation that is modestly stronger at 1-hour. Pearson's R calculated after testing for homoscedasticity in Graphpad Prism. (**k**) Endogenous Rosa-tdTom fluorescent micrographs of Ai14 mouse liver, lung, and spleen 24 hours post-intravenous (IV) administration of Spleen SORT LNPs carrying 0.1, 0.25, or 1.0 mg/kg Cre recombinase shows spleen-specific protein production. (**l,m**) Cre qPCR performed on liver, lung, and spleen 1-hour (**l**) or 24-hours (**m**) after administration of Liver SORT LNPs at various doses. (**n,o**) Linear regression analysis comparing tdTom signal vs. 1-hour qPCR (**n**) or 24-hour qPCR (**o**) shows linear correlation that is somewhat stronger at 1-hour. Pearson's R calculated after testing for homoscedasticity in Graphpad Prism. For qPCR; N=3 mice per dose, per time point, per formulation; 3 wells per measurement. For tdTom quantification used in correlation, N=3 image fields. Scale bars: 200 μ m.

Another problem with Figure 2 is that the title and description indicate that there is a dependence of the findings on time, however there are no statistical tests comparing the two data sets at the two time points.

Discussion: We appreciate your observation regarding the temporal interpretation in **Figure 2**. Our intent was to highlight a qualitative shift in biodistribution patterns between the 1-hour and 24-hour time points, particularly in relation to observed functional tropism. This shift was supported by correlation analyses, which revealed stronger alignment with expected tissue targeting at 1 hour compared to 24 hours. However, we agree that this data alone does not conclusively demonstrate time-dependence.

Action: We have updated the figure legend title to remove the claim of time dependence and revised the results section to reflect these changes and clarify the basis for our temporal interpretation includes subsequent NGS and kinetics experiments. We thank you for prompting this important clarification.

Other points on statistics:

1. Figure 3 and 4 are missing inferential statistics. Figure 5c does not describe the tests and comparisons. The authors should distinguish technical vs. biological replicates.

Discussion: We appreciate your attention to statistical robustness.

Action:

- We have added ANOVA results for **Figure 3** to the accompanying data file.
- **Figure 5c** caption has been updated to reflect that we performed ANOVA with Tukey's test for multiple comparisons.
- We have updated **Figure 4b** graphs to distinguish barcode replicates from animal replicates by plotting barcodes from each animal with a different symbol.

2. For Figure 2, data from micrographs are scaled by area, but data from PCR are normalized by actin, which should reflect cell number. The authors should describe how this may impact the correlation.

Discussion: We used Actin normalization to account for slight variations in RNA extraction, reverse transcription, and cDNA input between samples. The CT values for Actin between samples were typically within 1-2 PCR cycles for a given experiment.

The relationship between tdTom⁺ area fraction and tdTom⁺ cell number is expected to be approximately linear, provided that variation in individual cell size is small relative to overall biological variability and measurement noise. This assumption holds because the total positive area reflects the sum of individual tdTom⁺ cell areas. To mitigate confounding measurements from void regions—such as large blood vessels or the porous architecture of lung tissue—we applied a threshold and restricted quantification to regions of interest (ROIs) defined by DAPI signal. Under these conditions, the correlation between area-based and cell count-based metrics should remain stable, and calculated correlation coefficients are not expected to be significantly affected

Action: We have added a sentence to the results section describing the area measurement and underlying assumptions that reads as follows: “Here we used the tdTom⁺ area fraction of fluorescent micrographs in regions defined by DAPI signal to quantify the tdTom response, a measurement that assumes approximately linear scaling with the number of tdTom⁺ cells.”

3. When describing a graph, the nomenclature should be “y axis title” versus “x axis title.”

Discussion: We appreciate the stylistic suggestions.

Action: We have revised graph descriptions to follow the “y-axis vs. x-axis” format.

4. Bar graphs should not be used when plotting in log scale (doi: <https://doi.org/10.1101/2024.09.20.609464>)

Discussion: Thank you for highlighting this important visualization concern.

Action: In accordance with best practices outlined in the referenced preprint, we have revised all figures to use linear axes. Additionally, bar graphs previously shown on log-scaled plots have been replaced with XY scatter plots to more accurately represent the underlying data distribution and avoid potential misinterpretation. We appreciate the reviewer’s attention to clarity in data presentation.

Pharmacological mechanism of distribution. The most informative data shown is in Figure 3b, showing that within 30 seconds of injection, most of the material is no longer present in blood while high concentrations are found associated with tissues. This is highly unusual for colloiddally stable nanoparticles in the 100 nm size range. It is a well-known phenomenon that microparticles between 1-10 microns differentially distribute to the lung (for larger particles), liver, and spleen (PMID: 7391933, 7252811). Based on these well known effects, it seems likely that the LNPs upon injection rapidly form

occlusive micron-scale emboli that lodge in capillary beds. To my knowledge, there is no other known mechanism that can explain selective lung uptake within the time frame of two circulation passes in a mouse (15 s each). The authors should evaluate what happens to the administered LNPs within seconds after injection. I would expect that blood cells may be involved, possibly involving hemodynamic forces within capillaries, or this could be due to rapid LNP aggregate with blood components such as lipoproteins to form large, obstructive clusters. This is further consistent with the liver zonality, with preferential gene expression near the start of the capillary bed (periportal) which is expected to be the first region of occlusion for a microparticle.

Discussion: We appreciate your insight into classical pharmacology and agree that the rapid tissue association that we observed warrants further mechanistic investigation. We would like to gently clarify that, based on our measurements, the LNP signal in blood remains relatively high at the 30-second time point. It is not until approximately 5 minutes post-injection that we observe a marked decline in blood signal for liver- and spleen-targeting LNPs, consistent with rapid clearance or tissue uptake.

Interestingly, lung-targeting LNPs exhibit a more sustained blood signal over this early time window, suggesting differential circulation kinetics or retention mechanisms.

While our current study focuses on associations between barcode distribution and functional delivery, we are initiating follow-up experiments to explore early LNP fate.

Action: We have added a new supplementary figure that shows the distribution of LNPs loaded with Cy5-labeled mRNA 30 seconds and 5 minutes after injection.

To preserve the localization of LNPs in these very early observations we used the following procedure:

- Administered the LNPs, then immediately harvested the organs (~30s), or waited 5 minutes prior to harvesting
- Immediately flash froze tissues by placing them in OCT and partially submerging the block in liquid nitrogen
- Upon cryosectioning, we immediately mounted sections in mounting medium with DAPI and phalloidin-488 was added to visualize cell boundaries.
- Imaging was performed without fixation within 1 minute of obtaining each cryosection. All images shown were acquired within 5 minutes of obtaining each cryosection.

This process enabled us to preserve the localization of Cy5 mRNA labeled LNPs at a very early stage of delivery. We observed Cy5 puncta *within* cell boundaries, even at the 30s time point. The size of the puncta was consistent with endosomes. This experiment demonstrated that early organ accumulation *may* be driven by cellular uptake of LNPs, even on these short timescales. While we did not observe obvious occlusions in this experiment, we acknowledge that our data cannot rule out this mechanism. A future study should aim to explore this mechanism in greater detail. This new data has been added to **Supplementary Figure 4**.

Supplementary Figure 4. (g,h) Fluorescent micrographs showing the distribution of Cy5 mRNA loaded LNP signal 30 seconds and 5 minutes following LNP administration in the liver and lung. Live tissues were flash frozen, and all images were acquired within 5 minutes of cryosectioning to preserve LNP localization. Phalloidin 488 was added to mounting media to label filamentous actin (F-actin) and DAPI was used to label nuclei.

Thank you again for your detailed and thoughtful critique. We believe the proposed revisions and experiments significantly strengthen the manuscript's clarity, rigor, and mechanistic relevance.

References

1. Diehl KH, Hull R, Morton D, Pfister R, Rabemampianina Y, Smith D, Vidal JM, van de Vorstenbosch C; European Federation of Pharmaceutical Industries Association and European Centre for the Validation of Alternative Methods. A good practice guide to the administration of substances and removal of blood, including routes and volumes. *J Appl Toxicol.* 2001 Jan-Feb;21(1):15-23. doi: 10.1002/jat.727. PMID: 11180276.

Response to Referees

Reviewer #1 (Remarks to the Author):

The authors have provided a detailed and thoughtful response to Reviewer 4's comments and have made substantial revisions to the manuscript, figures, and supplementary data. Overall, the newly added experiments, expanded methodological descriptions, and improved statistical reporting have significantly strengthened the rigor and clarity of the study.

We thank the reviewer for their kind words and thoughtful review.

The only remaining comment relates to the second point concerning the distinction between this work and that of Mike Mitchell's lab (ref. 18). The current manuscript in fact advances barcoding-based screening beyond prior approaches by incorporating single-cell resolution and spatial information. These features open the door for future integration of barcoding technology with single-cell sequencing and spatial transcriptomics, which could greatly expand the impact and utility of the platform. Highlighting these possibilities more explicitly in the manuscript would further strengthen the case for novelty and address Reviewer 4's concerns regarding conceptual advance.

We thank the reviewer for highlighting this important distinction! We have revised the Discussion to emphasize that our platform establishes a framework to track LNPs with cellular resolution and spatial information, distinguishing it from prior barcoding-based screening approaches. The following sentence was added to the discussion: "We demonstrate that this approach provides a framework for multiplexed LNP tracking in defined cell populations, with the capacity to encode spatial and/or temporal information. This substantially expands the utility and potential applications of mRNA barcoding beyond its use as a primary screen."

Additionally, it may help reader understanding if the authors provide a reference when introducing the term "spatial flow cytometry." This will help orient readers who may be less familiar with this emerging concept. Overall, the manuscript has addressed Reviewer 4's critiques well, and emphasizing the points above would further clarify the platform's innovation and potential impact.

We thank the reviewer for this helpful suggestion. We have now included a reference when introducing the term "spatial flow cytometry" to better orient readers who may be less familiar with this emerging concept. This addition improves clarity and accessibility for a broad audience.

Reviewer #2 (Remarks to the Author):

The authors have addressed my major concerns.

1. The title of Part 4, "Barcoding streamlines new lipid discovery and formulation optimization," should be revised. I recommend deleting "new lipid discovery," as this study does not present any results related to the discovery or identification of new lipids.

We thank the reviewer for this helpful clarification. We have revised the title of Part 4 to remove the phrase "new lipid discovery," ensuring that the section more accurately reflects the scope of the study. The updated title now reads: "Barcoding streamlines formulation optimization."

2. I recommend that the authors include the gating strategy used in the liver metabolic zone study to facilitate reproducibility and make it easier for other researchers to follow the experimental workflow.

We thank the reviewer for this valuable suggestion. We have added a panel to **Supplementary Figure 6d** showing the gating strategy used in the liver metabolic zone study. This addition will facilitate reproducibility and provide clearer guidance for other researchers seeking to follow the experimental workflow.

Supplementary Figure 6. (d) Dot plots showing FACS gating strategy used to isolate zoned hepatocytes

We are grateful to the reviewer for their constructive comments, which have helped us improve the accuracy, clarity, and reproducibility of the manuscript.

Reviewer #5 (Remarks to the Author):

Thank you very much for your help!